# DATA ASSESSMENT FOR EMBODIED INTELLIGENCE

## ABSTRACT

In embodied intelligence, datasets play a pivotal role, serving as both a knowledge repository and a conduit for information transfer. The two most critical attributes of a dataset are the amount of information it provides and how easily this information can be learned by models. However, the multimodal nature of embodied data makes evaluating these properties particularly challenging. Prior work has largely focused on diversity, typically counting tasks and scenes or evaluating isolated modalities, which fails to provide a comprehensive picture of dataset diversity. On the other hand, the learnability of datasets has received little attention and is usually assessed post-hoc through model training—an expensive, time-consuming process that also lacks interpretability, offering little guidance on how to improve a dataset. In this work, we address both challenges by introducing two principled, data-driven tools. First, we construct a unified multimodal representation for each data and, based on it, propose diversity entropy, a continuous measure that characterizes the amount of information contained in a dataset. Second, we introduce the first interpretable, data-driven algorithm to efficiently quantify dataset learnability without training, enabling researchers to assess a dataset's learnability immediately upon its release. We validate our algorithm on both multiple simulated and real-world embodied datasets, demonstrating that it yields faithful, actionable insights, enabling researchers to jointly improve diversity and learnability. We hope this work provides a foundation for designing higher-quality datasets that advance the development of embodied intelligence.

## 1 INTRODUCTION

Recent advances in machine learning, particularly in computer vision and natural language processing, have been driven in large part by the scaling of both data and model sizes Radford et al. (2021); OpenAI (2024); Zhang et al. (2025). Empirical studies and scaling laws suggest that as the number of model parameters and the amount of training data increase, performance improvements and emergent generalization capabilities tend to follow predictable trends Kaplan et al. (2020), which highlight the importance of data in enabling models to gain more sophisticated knowledge.

In the domain of embodied intelligence, datasets play an analogous role as well Collaboration et al. (2025); Nezhurina et al. (2025). As the cornerstone of embodied intelligence, datasets have grown not only in size but also in diversity, evolving from small, task-specific collections Liu et al. (2023); Zhou et al. (2023b;c) to large-scale, general-purpose datasets such as Open-X Embodied Collaboration et al. (2025), Droid Khazatsky et al. (2024) and BridgeData Walke et al. (2023). These datasets support the training of powerful models, including Vision-Language-Action (VLA) models Team et al. (2024); Kim et al. (2024); Pertsch et al. (2025), which directly map observations to actions and aim to learn more generalizable embodied skills. Despite the rapid expansion of embodied datasets, two fundamental questions about their quality remain largely unaddressed: '*how much information does a dataset contain?*' and '*how easily can the information in a dataset be learned by models?*'.

Prior work has primarily focused on measuring dataset diversity, typically by counting the number of tasks or scenes, as illustrated in Figure 1, or by evaluating individual modalities in isolation Xing et al. (2025); Das et al. (2018); Xiao et al. (2025); Cheng et al. (2025). While these approaches offer a coarse estimate, they fail to capture the full richness of multimodal data, ignore redundancy between samples, and provide limited guidance on dataset design. Meanwhile, the learnability of datasets—the ease with which models can extract and generalize knowledge from the data—is typically quantified by training models on the dataset and measuring downstream performance, a process

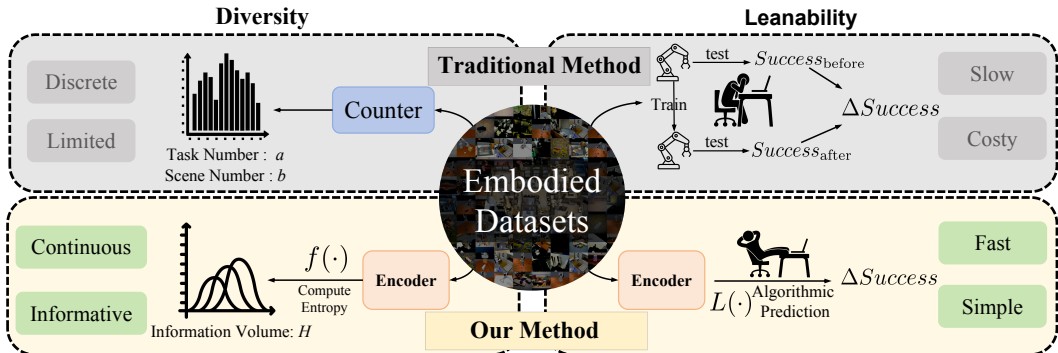

Figure 1: Comparison of traditional and our proposed approaches for embodied datasets. Top: traditional methods, where diversity is measured by counting tasks and scenes, and learnability is estimated via model training, observing the success rate improvement. Bottom: our approach, directly computing *diversity entropy* and *learnability* using principled, data-driven tools.

that is computationally expensive, time-consuming, and difficult to interpret. Such post-hoc evaluations do not provide actionable insights into how a dataset could be improved.

In this work, we introduce two principled, data-driven tools for embodied datasets. First, we propose *diversity entropy*, a continuous metric computed from a unified multimodal representation of each sample, which quantifies the information content and richness of a dataset. And we analyze 21 popular embodied datasets, encompassing over 800 GB of data. Second, we present the first interpretable, data-driven algorithm to efficiently estimate *dataset learnability* without requiring model training. This algorithm allows researchers to assess learnability immediately upon dataset release and provides insights into which aspects of the data facilitate or hinder learning. We validate its effectiveness through experiments on both multiple simulated and real-world embodied datasets, demonstrating that it reliably captures learnability patterns. Our framework lays the foundation for designing higher-quality datasets that advance embodied intelligence.

The main contributions of this paper can be summarized as follows: (1) We propose a unified representation method for embodied data which serves as the foundation for analyzing dataset properties. (2) Based on the unified representation, we introduce *diversity entropy*, a continuous metric that quantifies the information density and coverage of a dataset. (3) We present the first data-driven, training-free algorithm to efficiently estimate the learnability of a embodied dataset, enabling immediate evaluation and actionable insights for dataset improvement.

## 2 RELATED WORK

**Embodied datasets**   Recent progress in embodied intelligence has been driven by increasingly diverse datasets Ahn et al. (2022). The largest open-source collection, Open-X Embodiment Collaboration et al. (2025), contains roughly 1M trajectories, far below the trillion-scale corpora used in language modeling OpenAI (2025). To close this gap, recent work expands diversity along two directions: (1) cross-embodiment aggregation, unifying trajectories from heterogeneous robots into a shared vision–language–action space Brohan et al. (2023); Yang et al. (2025); and (2) high-fidelity synthetic data generation NVIDIA (2023), which samples challenging user–environment–task combinations in simulation, filters unsafe trajectories, and applies Sim2Real domain randomization Huber et al. (2023). These approaches increase data volume and nominal diversity but do not explicitly measure whether the data is informative or learnable for a given model.

**VLA Models.**   The rise of VLA models has turned embodied datasets from mere "validation sets" into the main training corpus Brohan et al. (2023). Early work extended Vision–Language models (VLMs) by appending actions as tokens Ahn et al. (2022), but suffered from coarse discretization. The RT series  Brohan et al. (2022)introduced an "action vocabulary" and fine-tuned VLM backbones, enabling zero-shot instruction following but still relying on ~130k trajectories from a single robot.  Recent cross-robot architectures such as OpenVLA  Kim et al. (2024) and CrossFormer Zhang & Yan (2023) align heterogeneous robot data in a shared latent space, enabling training on > 1M trajectories with 7B-parameter models. This shift redefines dataset construction, emphasizing diverse, multi-robot and multi-task coverage over dense sampling in a single environment.

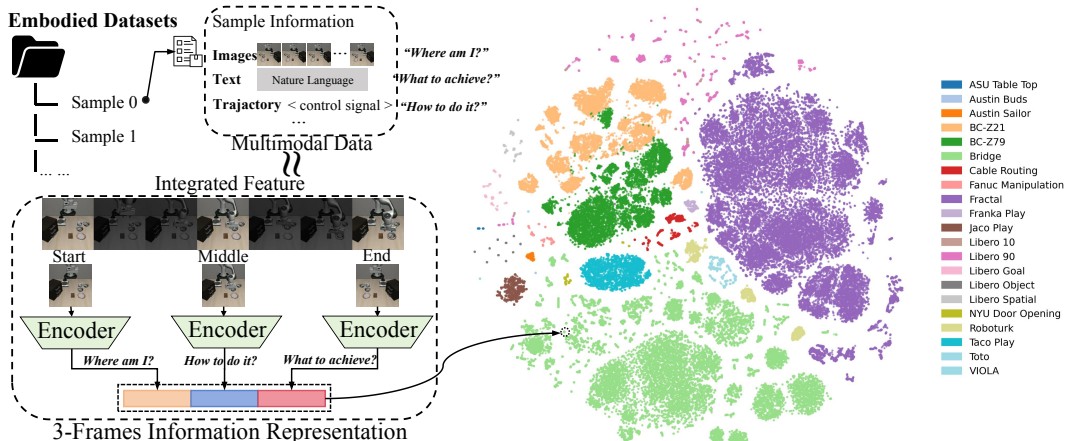

Figure 2: Visualization of the 3-Frames Information Representation for embodied datasets. Each sample is represented by a unified feature vector (left), and the distribution of all sample features across 21 popular embodied datasets is visualized using t-SNE (right).

## 3 METHODS

### 3.1 DIVERSITY

Embodied datasets are inherently multi-dimensional, typically including visual observations, action trajectories, and natural language instructions Khazatsky et al. (2024); Collaboration et al. (2025). These modalities respectively answer the questions *"Where am I?"*, *"How do I act?"*, and *"What task do I accomplish?"*. This rich, multi-faceted structure makes measuring dataset diversity substantially more challenging than in lower-modality or less structured datasets, such as one-dimensional purely textual datasets like WikiText Merity et al. (2016)and BookCorpus Singh (2024), or two-dimensional text-vision datasets like LAION Schuhmann et al. (2021).

**Prior approaches**    The most common way to measure diversity is simply to count the number of tasks and scenes Xiao et al. (2025); Li et al. (2023), but this approach essentially captures only discrete statistics, providing very coarse-grained information about diversity. Other works attempt to capture diversity by analyzing language instructions Collaboration et al. (2025); Li et al. (2025), while more recent methods embed instructions or visual scenes for diversity analysis Xing et al. (2025). However, these approaches have notable limitations: they often consider only a subset of modalities, typically ignoring the diversity introduced by different trajectories (e.g., demos with the same task in the same scene can still exhibit diversity due to different robot trajectories), and analyze each modality separately rather than providing a unified representation of dataset-level diversity. As a result, they fail to fully capture the rich, multi-dimensional structure inherent in embodied datasets.

**Unified Feature Representation**    To better represent multi-dimensional embodied data, we note that different modalities provide complementary views of the same underlying experience, and are thus temporally, spatially, and semantically consistent to some degree—much like how observing only the video frames can reveal both the scene and the task being performed Xu et al. (2021). Therefore, we hypothesize that the visual modality carries sufficient information to represent the entire sample. Based on this, we propose a 3-Frame information representation for each sample. As illustrated in Figure 2, we extract three representative frames corresponding to key stages—*first step* (*"where am I"*), *mid step* (*"how I act"*), and *last step* (*"what I accomplish"*)—and encode them using a vision-language alignment encoder such as clip Radford et al. (2021). The resulting frame embeddings are concatenated to obtain a unified feature vector $\mathbf{x}_i \in \mathbb{R}^D$ for each sample. The effectiveness of this approach is demonstrated in Section 4.3.

**Diversity Entropy**    Given these unified features, we define the Data Diversity Entropy $H_{\text{data}}$, which quantifies the effective diversity of the dataset $\mathcal{X} = \{\mathbf{x}_i\}_{i=1}^{|\mathcal{X}|} \subset \mathbb{R}^D$ in a purely representation-based manner. To estimate $H_{\text{data}}$, we adopt a Parzen window estimator Parzen (1962) with a Gaussian

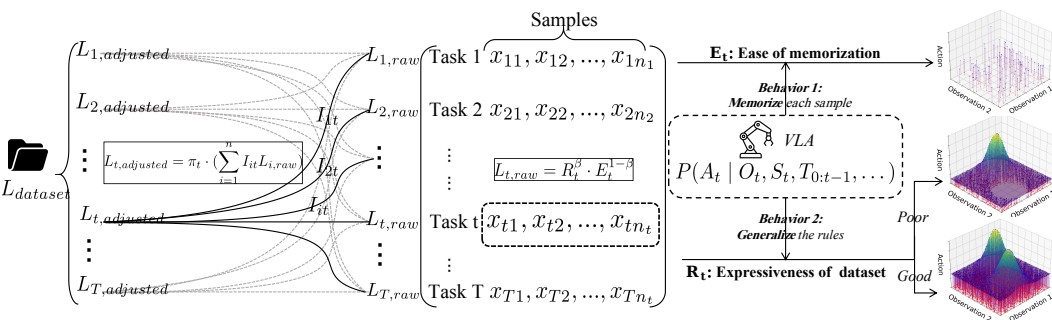

Figure 3: Overview of our learnability algorithm: model behaviors map to task attributes $E_t, R_t$ (right), compute raw learnability $L_{t,\text{raw}}$ for each tasks (middle), adjust for dataset influenced factors to get $L_{t,\text{transfer}}$ (left), and average across tasks for overall dataset learnability $L_{\text{dataset}}$.

kernel $K_\sigma(\cdot, \cdot)$, which serves as a kernel-based estimator of the differential entropy, computed as:

$$\hat{H}_{\text{data}} = -\frac{1}{|\mathcal{X}|} \sum_{\mathbf{x}_i \in \mathcal{X}} \log \left( \frac{1}{|\mathcal{X}|} \sum_{\mathbf{x}_j \in \mathcal{X}} K_\sigma(\mathbf{x}_i, \mathbf{x}_j) \right), \tag{1}$$

$H_{\text{data}}$ provides a principled, continuous measure of the intrinsic diversity of samples in the dataset. Formally, it quantifies the spread of the dataset in the high-dimensional feature space induced by the unified sample representation. As summarized in Table 1, $H_{\text{data}}$ responds systematically to both intra-task and inter-task variations: adding a new sample from a distinct task or one that lies far from existing samples increases $H_{\text{data}}$, reflecting an expansion of the overall feature coverage, whereas adding a new sample from the same task that is tightly clustered with existing samples decreases $H_{\text{data}}$, as redundant points concentrate the local distribution and reduce the effective diversity. Likewise, if the centers of different tasks move closer in feature space, $H_{\text{data}}$ decreases due to higher correlation between tasks, while greater separation between tasks increases $H_{\text{data}}$, indicating broader coverage of the feature space and more diverse task representations. Further details on the formulation of $H_{\text{data}}$, as well as a thorough discussion of this estimator, are provided in Appendix D.

## 3.2 LEARNABILITY

For a given dataset, it is important not only how much information the data contains, but also how effectively this information can be learned by a model—that is, the learnability $L_{\text{dataset}}$ of the dataset. In embodied datasets, learnability can be interpreted as the improvement in task success rate achieved by a VLA model Kim et al. (2024)after training on the dataset. However, due to the large scale and heterogeneous nature of embodied datasets Collaboration et al. (2025); Kalashnikov et al. (2018), conducting model-driven evaluations—training a model on the dataset and measuring task performance—is extremely time-consuming and resource-intensive Black et al. (2024); Team et al. (2024). Therefore, we aim to characterize the learnability of a dataset purely from the data itself.

**Model Behavior** We begin by considering how a model interacts with a dataset during learning. Conceptually, a model can be viewed as an ideal function attempting to capture the relationship between inputs and outputs. During training, the model may exhibit two distinct behaviors that reflect its tendency to overfit or generalize:

(1) Memorize each samples: corresponding to an overfitting tendency where the model simply memorizes each observed point, capturing its propensity for overfitting Arpit et al. (2017).

(2) Generalize underlying patterns: capturing statistical dependencies across samples, allowing the model to generalize beyond the observed points and uncover the latent structure of the dataset, reflecting its generalization ability Bengio et al. (2014).

**Data Properties** From the perspective of the data itself, the two behaviors of a model naturally correspond to two intrinsic properties of a dataset for a given task $t$. Together, these properties provide a principled understanding of a task's learnability, encompassing both the *ease of memorization* ($E_t$) and the *expressiveness* ($R_t$) that governs the potential for pattern extraction:

Table 1: Effects of sample changes on dataset diversity and each learnability factor.

| Sample Change | Affected Factors | Notes |
| --- | --- | --- |
| Add sample to a different task | $H_{\text{data}} \uparrow, \pi_t \downarrow, I_{it}$ | $I_{it}$ may shift with new task. |
| Add sample to task $t$, far from existing samples | $H_{\text{data}} \uparrow, \pi_t \uparrow, R_t \uparrow, E_t \downarrow, I_{it}$ | $I_{it}$ may decrease if $\mu_t$ moves. |
| Add sample to task $t$, tightly clustered | $H_{\text{data}} \downarrow, \pi_t \uparrow, R_t \downarrow, E_t \uparrow, I_{it}$ | $I_{it}$ may increase if $\mu_t$ shifts. |
| Tasks move closer in feature space | $H_{\text{data}} \downarrow, I_{it} \uparrow$ | $\mu_t$ shifts toward others |
| Tasks move farther apart in feature space | $H_{\text{task}} \uparrow, I_{it} \downarrow$ | $\mu_t$ shifts away |

(1) $E_t$ reflects how readily the samples of task $t$ can be memorized: tasks whose samples are highly similar in feature space can be learned by merely memorizing the trajectories, yielding high $E_t$. Conversely, if the samples of task $t$ exhibit considerable variation, even in the seen scenarios, memorization becomes more difficult, leading to lower $E_t$.

(2) $R_t$ captures the extent to which the samples of task $t$ reveal underlying patterns: if the samples exhibit sufficient variation, the model is more likely to uncover the underlying patterns necessary to accomplish the task, resulting in higher $R_t$. Conversely, tightly clustered samples may not expose the model to enough diversity, reducing $R_t$.

### 3.2.1 ALGORITHM MODELING

**Ease of Memorization Factor** $E_t$    The ease of memorization factor $E_t$ quantifies how easily a model can memorize the samples of task $t$. Higher $E_t$ indicates that the task is easier to overfit, whereas lower $E_t$ corresponds to tasks with more diverse samples that are harder to memorize. We account for two aspects: the ease of memorizing individual samples and the overall intra-task redundancy. First, the average operation steps $\bar{L}_t$ of all samples in task $t$ reflects the intrinsic difficulty of memorizing a single sample: longer sequences are harder to memorize, which we incorporate via the term $\log_{1p}^{-1}(\bar{L}_t)$. Second, the similarity between samples indicates redundancy within the task. Let each sample $\mathbf{x}_{t,i} \in \mathcal{X}_t \subset \mathbb{R}^D$, and let $K_\sigma(\mathbf{x}_{t,i}, \mathbf{x}_{t,j})$ be a Gaussian kernel measuring similarity between samples $i$ and $j$. The expectation over all sample pairs, $\mathbb{E}_{\mathbf{x}_{t,i}, \mathbf{x}_{t,j} \in \mathcal{X}_t}\big[K_\sigma(\mathbf{x}_{t,i}, \mathbf{x}_{t,j})\big]$, effectively counts the number of *distinct* samples: when all samples are identical, all pairwise similarities are 1, yielding $E_t \approx 1$, indicating that memorizing a single sample suffices; when all samples are dissimilar, the self-similarity dominates, and $E_t \approx 1/N$, reflecting that all $N$ samples must be memorized independently. Combining these terms, we define:

$$E_t = \log_{1p}^{-1}(\bar{L}_t) \cdot \mathbb{E}_{\mathbf{x}_{t,i}, \mathbf{x}_{t,j} \in \mathcal{X}_t}\big[K_\sigma(\mathbf{x}_{t,i}, \mathbf{x}_{t,j})\big]. \tag{2}$$

**Expressiveness Factor($R_t$)**    $R_t$ measures a task's potential for generalization to unseen samples. Intuitively, tasks with samples that span a wide range of parameters force the model to attend to multiple dimensions simultaneously and discover the relationships among parameters. In other words, high robustness occurs when task samples sufficiently cover the all possible parameter space. This ensures that the model is exposed to the full range of variability, enabling it to discover the underlying low-dimensional relationships and generalize to unseen scenarios. Formally, we define the expressiveness score for task $t$ as a combination of two complementary factors: directional coverage $H(X_t)$ and spatial coverage $C(X_t)$. The directional coverage $H(X_t)$ is quantified by the covariance spectrum entropy. Given a sample matrix $X_t \in \mathbb{R}^{N_t \times D}$, we compute its covariance matrix and obtain the eigenvalues $\lambda_1, \ldots, \lambda_D$, based on which the normalized entropy is calculated. This term captures how uniformly the variability is distributed across different dimensions, reflecting the isotropy of the data distribution in high-dimensional space. The spatial coverage $C(X_t)$ measures the effective spread of the samples in feature space. It is computed as the product of the number of samples $N_t$ and a nonlinearly scaled average pairwise distance $\bar{d}_t$, emphasizing both the density and overall spatial extent of the samples. Hence, the overall expressiveness for task $t$ is defined as:

$$R_t = \underbrace{-\sum_{i=1}^{D} \frac{\lambda_i}{\sum_{j=1}^{D} \lambda_j} \log \frac{\lambda_i}{\sum_{j=1}^{D} \lambda_j}}_{\text{Directional Coverage } H(X_t)} \times \underbrace{N_t \cdot \tanh\left(\frac{\bar{d}_t}{\sigma}\right)}_{\text{Spatial Coverage } C(X_t)}. \tag{3}$$

Multiplying these two components ensures that $R_t$ jointly reflects both high-dimensional variability and spatial coverage, providing a comprehensive measure of task expressiveness.

| Dataset | SRCC↑ | KRCC↑ | PLCC↑ |
|---------|-------|-------|-------|
| Libero-Goal | 0.6159 | 0.4319 | 0.7479 |
| Libero-Object | 0.3303 | 0.2300 | 0.1975 |
| Libero-Spatial | 0.2553 | 0.2247 | 0.3678 |
| Libero-10 | 0.6848 | 0.4667 | 0.6743 |
| Libero-Goal+10 | 0.7622 | 0.5990 | 0.7955 |
| All | 0.7974 | 0.6033 | 0.7966 |

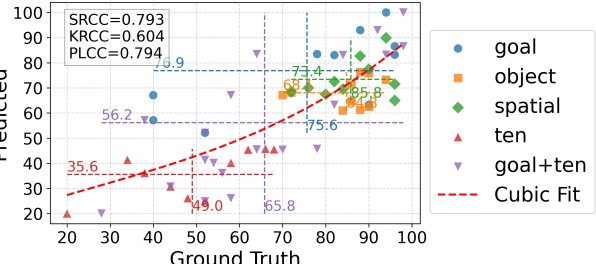

Figure 4: Validation on the simulated dataset, showing results for each subset and the full dataset (left) and a scatter plot of predicted vs. ground-truth scores with dataset-level reference lines (right).

*Task: Pick up the red cube    Initial Success Rate: 2%*

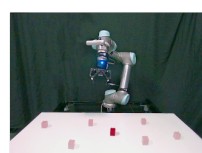
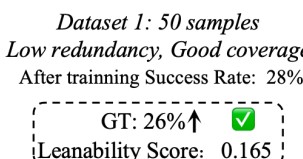
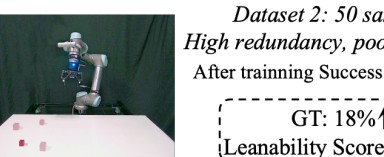

Figure 5: Validation on two real-world datasets collected with a UR5 robot. Our method correctly reflects the relative learnability of the datasets, demonstrating its effectiveness in real-world settings.

**Learnability**    Combining the two intra-task factors, the raw learnability of task $t$ is:

$$L_{t,\text{raw}} = R_t^{\beta} \cdot E_t^{1-\beta}, \quad \beta \in [0,1], \tag{4}$$

where $\beta$ controls the relative weighting between the expressiveness score $R_t$ and the memory ease factor $E_t$. After computing the raw task learnability $L_{t,\text{raw}}$, we account for both inter-task transfer and the relative representation of each task in the dataset, as shown in Figure 3. The influence of task $i$ on task $t$ is quantified by $I_{it} = K_{\sigma}(\mu_i, \mu_t)$, where $\mu_t$ is the center of task $t$'s samples and $K_{\sigma}(\cdot, \cdot)$ is a Gaussian kernel capturing task similarity. The relative representation is reflected by a sample proportion factor $\pi_t = \tanh(|X_t|/(\sum_t N_t \cdot \sigma_{\text{model}}))$, where $\sigma_{\text{model}}$ controls the scaling according to the model's capacity. Combining these two aspects, the adjusted task learnability is defined as:

$$L_{t,\text{adjusted}} = \pi_t \cdot \left( \sum_i I_{it} L_{i,raw} \right) = \tanh \left( \frac{|X_t|}{\sum_t N_t \cdot \sigma_{\text{model}}} \right) \cdot \left( \sum_i K_{\sigma}(\mu_i, \mu_t) L_{i,raw} \right), \tag{5}$$

and the dataset learnability is the mean adjusted task learnability: $L_{\text{dataset}} = \frac{1}{|T|} \sum_{t \in T} L_{t,\text{adjusted}}$.

$L_{t,\text{adjusted}}$ provides a comprehensive measure of dataset-level learnability by integrating intra-task properties, inter-task transfer effects, and the relative representation of each task in the dataset. As summarized in Table 1, adding a new sample to a different task reduces the relative task prior $\pi_t$ of task $t$, since its proportion in the overall dataset decreases, while $R_t$ and $E_t$ for task $t$ remain unchanged but the inter-task similarity $I_{it}$ may shift slightly if the added point moves the task center $\mu_i$. Adding a sample to task $t$ that lies far from existing samples simultaneously increases $\pi_t$ and raises $R_t$, as the new point expands the local coverage and increases neighborhood entropy; the reduced local density makes memorization more difficult, resulting in a lower $E_t$, and may slightly decrease $I_{it}$ by increasing the separation from other tasks. Conversely, adding a tightly clustered sample to task $t$ still increases $\pi_t$ but reduces $R_t$ by lowering local entropy, while $E_t$ rises due to the higher density that facilitates memorization and $I_{it}$ may increase if the new sample moves $\mu_t$ closer to $\mu_i$. Further details on the formulation and explanation of $L_{\text{dataset}}$ are provided in Appendix E.

## 4  EXPERIMENT

### 4.1  EXPERIMENTAL VERIFICATION FOR LEARNABILITY

**Experiment Setup**    Obtaining reliable ground-truth measurements of dataset learnability faces two main challenges. Training a large embodied model from scratch or pretraining on large-scale datasets is prohibitively time- and resource-intensive Kim et al. (2024), and exact replication of real-world task scenarios to evaluate is often infeasible (More details on the challenges can be found

Table 2: Embodied datasets summary on: number of samples, size (GB), image **res**olution, camera views number, action **dim**ensionality, normalised standard deviations of five low-level features: **Lum**inance, **S**patial **I**nfomation., **Contr**ast, **Chrom**inance, **Blur**, and the estimated entropy $\hat{H}_{\text{data}}$.

| Dataset name | #Samples | Size | Res. | Views | Dim. | Lum. | S.I. | Contr. | Chrom | Blur | $\hat{H}_{\text{data}}$ |
|---|---|---|---|---|---|---|---|---|---|---|---|
| ASU Table Top Zhou et al. (2023b) | 110 | 0.72 | 224p | 1 | 7 | 0.0151 | 0.0195 | 0.0123 | 0.0114 | 0.0051 | 4.6986 |
| Austin Buds Zhu et al. (2022b) | 50 | 1.49 | 128p | 2 | 7 | 0.0061 | 0.0032 | 0.0215 | 0.0034 | 0.0118 | 3.9120 |
| Franka Play Cui et al. (2022) | 365 | 5.18 | 128p | 2 | 15 | 0.0180 | 0.0115 | 0.0210 | 0.0359 | 0.0439 | 5.8998 |
| BC-Z79 Jang et al. (2021) | 9,106 | 16.43 | 171p | 1 | 10 | 0.0318 | 0.0232 | 0.0408 | 0.0388 | 0.0114 | 9.1100 |
| BC-Z21 Jang et al. (2021) | 9,746 | 13.63 | 171p | 1 | 10 | 0.0319 | 0.0318 | 0.0529 | 0.0330 | 0.0160 | 9.1845 |
| Jaco Play Dass et al. (2023) | 976 | 9.24 | 224p | 2 | 7 | 0.0132 | 0.0117 | 0.0285 | 0.0630 | 0.0216 | 6.8832 |
| Berkeley Cable Routing Luo et al. (2023) | 1,482 | 4.67 | 128p | 4 | 7 | 0.0687 | 0.0612 | 0.0850 | 0.0277 | 0.0401 | 7.2464 |
| Austin Sailor Nasiriany et al. (2022) | 240 | 18.85 | 128p | 2 | 7 | 0.0105 | 0.0086 | 0.0157 | 0.0312 | 0.0444 | 5.4804 |
| Roboturk Mandlekar et al. (2019) | 1,786 | 45.39 | 480p | 1 | 7 | 0.0596 | 0.0446 | 0.0738 | 0.2453 | 0.0404 | 7.4877 |
| Libero 10 Liu et al. (2023) | 500 | 13.73 | 224p | 2 | 7 | 0.1081 | 0.1208 | 0.1133 | 0.0596 | 0.0820 | 6.1815 |
| Libero 90 Liu et al. (2023) | 4,500 | 66.69 | 224p | 2 | 7 | 0.1003 | 0.1401 | 0.1311 | 0.0743 | 0.0684 | 8.3448 |
| Libero Goal Liu et al. (2023) | 500 | 6.38 | 224p | 2 | 7 | 0.0015 | 0.0015 | 0.0132 | 0.0010 | 0.0047 | 6.1164 |
| Libero Object Liu et al. (2023) | 500 | 7.45 | 224p | 2 | 7 | 0.0014 | 0.0034 | 0.0171 | 0.0105 | 0.0119 | 6.1227 |
| Libero Spatial Liu et al. (2023) | 500 | 6.75 | 224p | 2 | 7 | 0.0045 | 0.0052 | 0.0210 | 0.0019 | 0.0193 | 6.1701 |
| NYU Door Opening Pari et al. (2021) | 435 | 7.12 | 720p | 1 | 7 | 0.0331 | 0.0627 | 0.1761 | 0.0211 | 0.0309 | 5.2685 |
| Taco Play Rosete-Beas et al. (2022) | 3,242 | 47.77 | 150p | 1 | 15 | 0.0100 | 0.0116 | 0.0414 | 0.0136 | 0.0207 | 7.4877 |
| Toto Play Zhou et al. (2023a) | 902 | 13.88 | 480p | 1 | 7 | 0.0124 | 0.0172 | 0.0365 | 0.0657 | 0.0190 | 6.8032 |
| Viola Zhu et al. (2022a) | 300 | 10.40 | 224p | 2 | 7 | 0.0863 | 0.0821 | 0.0091 | 0.1404 | 0.0841 | 5.6978 |
| Fanuc Manipulation Zhu et al. (2023) | 415 | 8.85 | 224p | 2 | 6 | 0.0145 | 0.0845 | 0.1719 | 0.1684 | 0.0275 | 6.0016 |
| Fractal Brohan et al. (2022) | 87,212 | 111.38 | 320p | 1 | 10 | 0.0294 | 0.0418 | 0.0903 | 0.0575 | 0.0426 | 11.372 |
| Bridge Walke et al. (2023) | 28,935 | 387.5 | 256p | 4 | 7 | 0.0396 | 0.0612 | 0.0967 | 0.2124 | 0.0442 | 9.9118 |

in Appendix G).To address this, we adopt a fine-tuning paradigm on datasets that allow task-level validation, efficiently measuring the effect of each dataset on model performance. Experiments are conducted on seven datasets: five simulated RoboSuite datasets Zhu et al. (2020) (Libero-object, Libero-spatial, Libero-goal, Libero-10, and the combined Libero-goal+Libero-10 Liu et al. (2023)) and two real-world datasets collected in our lab (see Appendix H for details). For preliminary validation, we fix the model to OpenVLA-7B Kim et al. (2024), chosen for its strong embodied reasoning capabilities and wide adoption, allowing us to focus on dataset-level learnability. OpenVLA-7B is fine-tuned using LoRA adapters with rank 32 for 35,000 steps, global batch size 32, gradient accumulation 1, and learning rate $5 \times 10^{-4}$ on 4×H200 GPUs in distributed data-parallel mode via torchrun. After fine-tuning, we evaluate each task both before and after training, running 50 episodes per task; the increase in success rate serves as the ground-truth task-level learnability. This yields 60 task-level points, used to compute Spearman's rank correlation coefficient (SRCC), Kendall's rank correlation coefficient (KRCC), and Pearson's linear correlation coefficient (PLCC) between predicted and ground-truth scores. Metric hyperparameters are set as $\beta = 0.5$, $\sigma_{\text{model}} = 0.02$, task-internal bandwidths $\sigma_t = 0.001$, and inter-task similarity $\sigma_{\text{center}} = 0.01$.

**Simulated Datasets Validation Result** The table in Figure 4 reports the correlation between our predicted task-level learnability scores and the ground-truth scores on five Libero datasets. Our method achieves strong overall correlation across all tasks, with SRCC = 0.7974, KRCC = 0.6033, and PLCC = 0.7966, demonstrating the effectiveness of our approach in capturing the relative difficulty of tasks. At the per-dataset level, our method correctly preserves the relative ordering of most datasets, mis-ranking only Libero-Goal. This is also evident from the predicted vs. ground-truth scatter plot in Figure 4, where the points lie close to the diagonal, indicating strong agreement. In addition, we observe relatively lower correlations for Libero-Object and Libero-Spatial. This may arise partly from inherent task characteristics: Libero-Object exhibits variations in object identity, while Libero-Spatial involves differences in spatial positions. In both cases, the differences between tasks are naturally small, as reflected by the low task-to-task variance in Figure 4, making it challenging even for ground-truth scores to clearly distinguish them. More detailed numerical results, including the raw task-level scores, can be found in Appendix F.

**Real-World Datasets Validation Result** We further validate our metric on two real-world datasets collected in our lab using a UR5 robot performing a "pick up the red cube" task. Each dataset contains 50 expert demonstrations. Dataset 1 was collected to maximize coverage and minimize redundancy, while Dataset 2 was collected with high redundancy and lower coverage. As illustrated in Figure 5, before fine-tuning, the baseline OpenVLA model achieved a 2% success rate. After fine-tuning with LoRA, the success rate increased by 26% on Dataset 1 and 18% on Dataset 2. Our predicted learnability scores were 0.165 and 0.158 for Dataset 1 and Dataset 2, respectively, correctly reflecting that Dataset 1 is more learnable for improving model performance. These results demonstrate that our method generalizes well beyond simulated environments and is effective for characterizing the quality of real-world datasets.

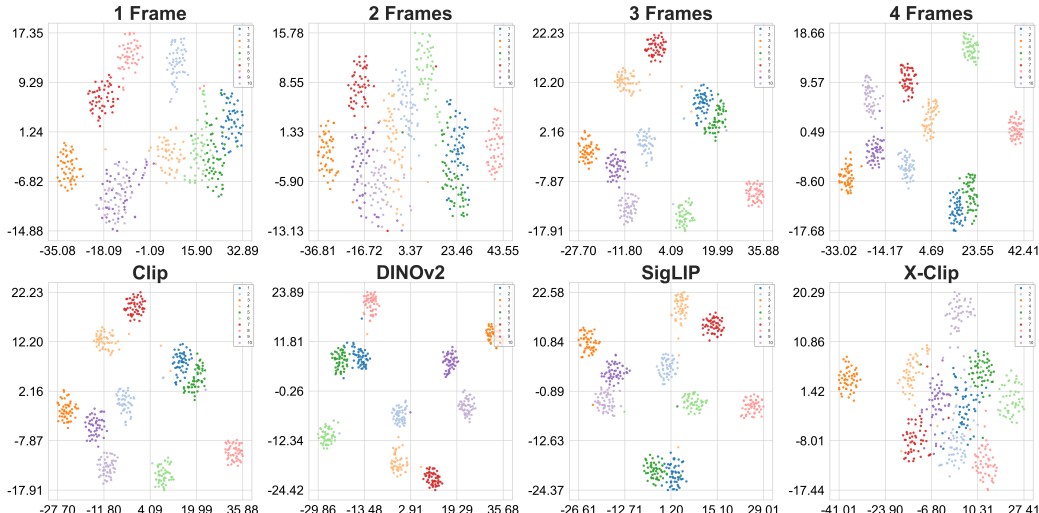

Figure 6: Ablation study of different frames and model choices for multimodle representation.

## 4.2 EMBODIED DATASET DIVERSITY ANALYSIS

**Experiment Setup** We analyzed 21 embodied datasets, including *ASU Table Top* Zhou et al. (2023b), *Austin Buds* Zhu et al. (2022b), *Franka Play* Cui et al. (2022), *BC-Z* Jang et al. (2021), *Jaco Play* Dass et al. (2023), *Berkeley Cable Routing* Luo et al. (2023), *Austin Sailor* Nasiriany et al. (2022), *Roboturk* Mandlekar et al. (2019), *Libero* Liu et al. (2023), *NYU Door Opening* Pari et al. (2021), *Taco Play* Rosete-Beas et al. (2022), *Toto* Zhou et al. (2023a), *Viola* Zhu et al. (2022a), *Fanuc Manipulation* Zhu et al. (2023), *Fractal* Brohan et al. (2022), and *Bridge* Walke et al. (2023), covering tasks such as tabletop manipulation, robot grasping, and simulated interactions. As summarized in Table 2, these datasets contain between 50 and 87,212 samples, totaling approximately 800 GB, with diverse image resolutions, camera views, and action dimensionalities. To quantify low-level feature diversity, we computed normalized standard deviations for luminance, spatial information, contrast, chrominance, and blur (details in Appendix L), as well as the overall diversity entropy $\hat{H}_{\text{data}}$, using a kernel density estimation approach with a Gaussian bandwidth of 0.1 on a single H200 GPU. Sample features were extracted with CLIP using a 3-frame representation, and the distributions across datasets are visualized using t-SNE in Figure 2. The resulting diversity entropy values, reported in Table 2, provide a comprehensive overview of dataset complexity and variability.

**Results and Analysis** We observe several interesting patterns from the results. Among all datasets, *Fractal* exhibits the highest diversity entropy, reaching 11.3718, indicating a rich coverage of task and observation space, while *Austin Buds* has the lowest diversity entropy at only 3.9120, reflecting limited variability. Overall, larger datasets tend to show higher diversity entropy, but sample count alone is not determinative: for example, *NYU Door Opening* contains 435 samples—more than *Viola*, *NYU Franka Play*, and *Austin Sailor*—yet achieves a lower diversity entropy of 5.2685, suggesting substantial redundancy. Similarly, the four Libero datasets all contain 500 samples yet yield entropy values from 6.1164 to 6.1815. The low-level visual statistics show minimal variation across datasets—likely due to shared camera setups, compression, and resolution—thus offering limited signal for distinguishing dataset diversity. Taken together, these observations indicate that neither sample count nor low-level feature variation fully determines dataset diversity; instead, true dataset richness depends on higher-level semantic and task-related differences, which our entropy-based metric effectively captures (see Appendix L for details). Hence, while increasing data volume can still improve diversity, focusing solely on scaling may not be the most efficient use of resources given the cost of data collection. A more economical strategy is to complement dataset growth with data curation and diversity-aware generation—producing data that is richer and less repetitive—thus increasing the *information conversion rate* per sample and enabling even moderately sized datasets to exhibit stronger scaling behavior with respect to learnability.

## 4.3 COMPARATIVE AND ABLATION EXPERIMENTS

In this section, we conduct two sets of experiments. First, we perform comparative studies to justify our design choices of adopting a 3-frame representation and CLIP backbone. Second, we conduct

| Frames | Accuracy↑ | Density↑ | Stability↑ | Time (s) |
|---|---|---|---|---|
| 1 | 0.200 | 0.1650 | 0.828 | 34.00 |
| 2 | 0.200 | 0.1740 | 0.924 | 56.84 |
| 3 | 0.868 | 0.143 | 0.978 | 74.27 |
| 4 | 0.898 | 0.1341 | 0.998 | 97.60 |

| Frames | Accuracy↑ | Density↑ | Stability↑ | Time (s) |
|---|---|---|---|---|
| Clip | 0.868 | 0.143 | 0.978 | 74.27 |
| DINOv2 | 0.892 | 0.160 | 0.986 | 179.59 |
| SigLIP | 0.700 | 0.112 | 0.982 | 156.19 |
| X-Clip | 0.200 | 0.129 | 0.920 | 513.60 |

Table 3: Comparison of Accuracy, Density (Silhouette), Stability (5-fold) performance for different frame counts (on libero-object dataset, left) and models (on libero-spatial dataset, right).

an ablation study on the transfer block to verify its effectiveness in transforming $L_{t,\mathrm{raw}}$ into the final learnability score $L_{t,\mathrm{transfer}}$. Further experiments are provided in Appendix I.

**Frame Number Comparison**   Our goal in selecting the frame sampling strategy and vision–language backbone is to ensure that the resulting feature representations are both discriminative and efficient. Ideally, the learned feature space should cluster samples by task and scene (coarse-grained distinction), while still separating different trajectories within the same task (fine-grained distinction). To this end, we first investigate the effect of the number of frames used for CLIP-based feature extraction on the Libero-object dataset. For each strategy (1-frame: first frame only; 2-frame: first and last; 3+ frames: first, last, and uniformly sampled middle frames), we extract features and visualize them in the latent space to inspect whether samples cluster according to task. We further evaluate the quality of clustering using an unsupervised KMeans classifier, measuring classification accuracy, Silhouette score, and 5-fold cross-validation accuracy. As shown in Figure 6, when using three or more frames, sample points clearly cluster by task, and 5-fold accuracy exceeds 95%. While using four frames yields slightly better performance, it also increases computational cost by over 30%, making it less suitable for large-scale dataset analysis. We therefore adopt the 3-frame strategy as the default to balance representation richness and efficiency.

**Vision Encoder Comparison**   We also compare several vision–language aligned encoders for feature extraction, including Clip Radford et al. (2021), DINOv2 Oquab et al. (2024), SigLIP Zhai et al. (2023), and video model X-Clip Ma et al. (2022). As shown in Table 3, Clip, DINOv2, and SigLIP all yield high-quality clustering results, while X-Clip struggles both in accuracy and efficiency, achieving only around 0.20 in classification accuracy and requiring the longest inference time among all models. Although DINOv2 attains slightly higher classification accuracy than Clip, its computation time is more than double, which makes it less practical for large-scale experiments. Considering both performance and efficiency, we adopt Clip as the default vision encoder, as it offers the best balance between accuracy and computational cost.

$L_{t,\mathbf{raw}}$ **VS** $L_{t,\mathbf{transfer}}$   We evaluate the impact of the task transfer block, as shown in Table 4. The experiments are conducted on the same five Libero datasets used in the previous evaluation, comparing the learnability pre-

| Model | SRCC↑ | KRCC↑ | PLCC↑ | Mean↑ |
|---|---|---|---|---|
| $L_{t,\mathrm{raw}}$ | 0.7925 | 0.6038 | 0.7944 | 0.7302 |
| $L_{t,\mathrm{transfer}}$ | 0.7974 | 0.6033 | 0.7966 | 0.7324 |

Table 4: Ablation for the transfer block

dictions using the original features $L_{t,\mathrm{raw}}$ and those processed through the transfer block $L_{t,\mathrm{transfer}}$. Adding the transfer block leads to modest improvements across all correlation metrics. Specifically, SRCC increases from 0.7925 to 0.7974, PLCC rises from 0.7944 to 0.7966, and the overall mean score improves from 0.7302 to 0.7324. Although the gains are moderate, these results show that the transfer block enhances the agreement between predicted and ground-truth learnability.

## 5 CONCLUSION

In this work, we presented two principled, data-driven tools for assessing embodied datasets. First, we introduced *diversity entropy*, characterizing the information richness of a dataset. Experiments on 21 large-scale embodied datasets, we observe that simply scaling the number of episodes yields diminishing returns in information gain. An economical path forward is to raise the information conversion rate per sample—curating or generating frames that carry richer, less repetitive signals. Second, we developed the first interpretable algorithm to efficiently estimate dataset *learnability* without model training. Experiments on several simulated and real-world datasets demonstrate the effectiveness of our algorithm, indicating that poor embodied capabilities are closely related to data itself. We hope this initial attempt will inspire the community to prioritize quality and quantity improvements in embodied datasets, thereby advancing the evolution of embodied intelligence models.

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

# A  AUTHOR STATEMENT

**Ethics Statement:**  This work does not involve any sensitive personal data, private information, or high-risk deployment scenarios. Our evaluation relies solely on publicly available embodied datasets and synthetic environments, and no human subjects were involved beyond informal pilot studies that do not require IRB approval. We believe that releasing reproducible metrics for diversity and learnability can have a positive impact on the community by enabling fairer, more systematic dataset assessment and curation.

**LLM Usage:**  Large-language models were employed exclusively for formula verification, language polishing, and auxiliary code scaffolding. No scientific conclusions, experimental designs, or core contributions were generated by LLM.

**Reproducibility:**  To foster reproducibility and facilitate adoption, we will release not only the code and data processing pipelines but also provide a well-packaged, user-friendly toolkit. This toolkit will allow researchers to compute both diversity entropy and learnability metrics with a single command by specifying the dataset path, ensuring that our contributions are easily reusable and extensible by the community.

# B  DISCUSSION

This paper advocates for greater attention to the information content and learnability of embodied datasets. We propose a preliminary, data-driven approach to investigate how data characteristics influence dataset information and learning efficiency. Experiments were conducted on a small set of real and synthetic datasets via fine-tuning, with OpenVLA as the sole testbed; different model preferences may introduce minor biases. Our method focuses exclusively on datasets and data—no model architectures are modified. Broader applicability and limitations of our approach remain to be explored in future work. We hope this initial attempt will inspire the community to prioritize quality and quantity improvements in embodied datasets, thereby advancing the evolution of embodied intelligence models.

# C  FUTURE WORK

Our framework opens several promising directions for future research. First, the proposed *diversity entropy* metric can serve as a principled standard for quantifying dataset scale in future large-scale VLA datasets. Future work could explore more expressive sample-level representations, such as increasing the number of frames per demonstration, integrating additional modalities, or employing stronger encoders to capture richer information. Second, the modeling of environment and task difficulty in our learnability algorithm can be further refined. For instance, rather than relying solely on task length, future extensions could incorporate trajectory smoothness, number of critical states, or motion complexity as additional indicators of execution difficulty. Finally, an exciting direction is to embed learnability estimation into closed-loop data collection pipelines, enabling adaptive dataset curation where both diversity and learnability are jointly optimized, ultimately accelerating model improvement and ensuring efficient data usage.

# D  MORE DETAILS ABOUT $H_{\text{DATA}}$

In this section, we provide a detailed explanation of how we compute the dataset diversity measure $H_{\text{data}}$ using a non-parametric kernel-based estimator.

## D.1  PARZEN WINDOW DENSITY ESTIMATE

We first represent the dataset as a set of unified multimodal feature vectors $\mathcal{X} = \{\mathbf{x}_i \in \mathbb{R}^D\}_{i=1}^{|\mathcal{X}|}$, where each $\mathbf{x}_i$ is the feature embedding of one sample. To estimate the underlying probability density $p(\mathbf{x})$ of the data distribution, we adopt the Parzen window (or kernel density) estimator,

which places a smooth kernel around each observed point:

$$\hat{p}(\mathbf{x}) = \frac{1}{|\mathcal{X}|} \sum_{j=1}^{|\mathcal{X}|} K_\sigma(\mathbf{x}, \mathbf{x}_j).$$

Here, $K_\sigma(\mathbf{x}, \mathbf{x}_j)$ is a Gaussian kernel measuring the similarity between $\mathbf{x}$ and $\mathbf{x}_j$:

$$K_\sigma(\mathbf{x}, \mathbf{x}_j) = \frac{1}{(2\pi\sigma^2)^{D/2}} \exp\left(-\frac{\|\mathbf{x} - \mathbf{x}_j\|^2}{2\sigma^2}\right),$$

where $\sigma > 0$ is the bandwidth controlling the smoothness of the density estimate: a small $\sigma$ captures fine-grained variations but may overfit, whereas a large $\sigma$ produces a smoother estimate but may oversmooth local structure.

This approach gives a non-parametric, sample-based estimate of the density without assuming any parametric form (e.g., Gaussian mixture). It is particularly suitable for high-dimensional, multi-modal datasets where the true distribution may be complex.

## D.2 KERNEL ENTROPY ESTIMATOR

With $\hat{p}(\mathbf{x})$ in hand, we can compute the (differential) Shannon entropy of the dataset, which quantifies the average "uncertainty" or "spread" of samples in the feature space:

$$H(p) = -\mathbb{E}_{\mathbf{x} \sim p}\big[\log p(\mathbf{x})\big].$$

Replacing $p(\mathbf{x})$ by its Parzen window estimate $\hat{p}(\mathbf{x})$ and approximating the expectation by an empirical average over the dataset yields the kernel entropy estimator:

$$\hat{H}_{\text{data}} = -\frac{1}{|\mathcal{X}|} \sum_{i=1}^{|\mathcal{X}|} \log \hat{p}(\mathbf{x}_i).$$

Intuitively, if data points are densely packed, $\hat{p}(\mathbf{x}_i)$ will be large, resulting in a smaller entropy. If data points are well spread out and diverse, $\hat{p}(\mathbf{x}_i)$ becomes smaller, yielding a larger entropy. Thus, $\hat{H}_{\text{data}}$ serves as a continuous and differentiable measure of dataset diversity.

In practice, we tune the kernel bandwidth $\sigma$ to balance bias and variance of the estimate (e.g., via Silverman's rule-of-thumb or cross-validation). This ensures that the diversity score reflects the intrinsic distribution of the dataset rather than artifacts of the estimator.

## D.3 BOUNDS AND EXTREMAL CASES OF $\hat{H}_{\text{DATA}}$

We provide explicit upper and lower bounds for the Parzen window entropy estimator and discuss the extremal cases where these bounds are attained.

**Setup**    Recall that for a dataset and the Parzen density estimate: $\mathcal{X} = \{\mathbf{x}_i\}_{i=1}^{n}$ and Gaussian kernel

$$K_\sigma(\mathbf{x}, \mathbf{x}') = \frac{1}{(2\pi\sigma^2)^{D/2}} \exp\left(-\frac{\|\mathbf{x} - \mathbf{x}'\|^2}{2\sigma^2}\right), \hat{p}(\mathbf{x}_i) = \frac{1}{n} \sum_{j=1}^{n} K_\sigma(\mathbf{x}_i, \mathbf{x}_j).$$

Denote the zero-distance kernel value as

$$K(0) \equiv K_\sigma(\mathbf{x}, \mathbf{x}) = \frac{1}{(2\pi\sigma^2)^{D/2}}.$$

**Bounding the Density**    For each $i$, by the non-negativity of $K_\sigma(\cdot, \cdot)$ we have

$$\frac{1}{n} K(0) \;\leq\; \hat{p}(\mathbf{x}_i) \;\leq\; K(0).$$

The lower bound is attained when all cross-kernel terms $K_\sigma(\mathbf{x}_i, \mathbf{x}_j)$ for $j \neq i$ vanish (i.e., samples are mutually far apart compared to the kernel bandwidth), and the upper bound is attained when all samples coincide.

**Entropy Bounds**  Substituting the above inequality into

$$\hat{H}_{\text{data}} = -\frac{1}{n} \sum_{i=1}^{n} \log \hat{p}(\mathbf{x}_i),$$

we obtain

$$-\log K(0) \;\leq\; \hat{H}_{\text{data}} \;\leq\; -\log\!\Big(\frac{K(0)}{n}\Big) = -\log K(0) + \log n.$$

Expanding $K(0)$ leads to an explicit closed-form:

$$\boxed{\frac{D}{2} \log(2\pi\sigma^2) \;\leq\; \hat{H}_{\text{data}} \;\leq\; \frac{D}{2} \log(2\pi\sigma^2) + \log n.}$$

**Extremal Cases**

- **Lower bound (minimal entropy):** $\hat{H}_{\text{data}} = \frac{D}{2} \log(2\pi\sigma^2)$ is achieved when $\mathbf{x}_1 = \cdots = \mathbf{x}_n$ (all samples coincide). This corresponds to a *zero-diversity* dataset.

- **Upper bound (maximal entropy):** $\hat{H}_{\text{data}} = \frac{D}{2} \log(2\pi\sigma^2) + \log n$ is approached when all samples are mutually far apart (relative to $\sigma$), so that cross-kernel contributions are negligible. This represents the case of maximal coverage of the feature space.

**Interpretation**  These closed-form bounds show that:

1. Increasing $\sigma$ raises both bounds, as a wider kernel produces lower density values and hence higher entropy.

2. Higher feature dimension $D$ linearly increases both bounds, reflecting the larger volume of the Gaussian kernel in higher dimensions.

3. Increasing sample size $n$ only affects the *upper* bound via $\log n$, meaning that entropy can increase at most logarithmically with more mutually distant samples.

In practice, these bounds serve as a useful reference scale: if the estimated $\hat{H}_{\text{data}}$ is close to the lower bound, the dataset contains significant redundancy; if it approaches the upper bound, the dataset covers the feature space broadly with little overlap.

### D.4  BEHAVIOR ANALYSIS

| Sample Change | Effect on $H_{\text{task}}$ | Explanation |
|---|---|---|
| Add new task sample far from existing samples | $H_{\text{task}} \uparrow$ | New task increases global feature diversity |
| Add same-task sample tightly clustered | $H_{\text{task}} \downarrow$ | Redundant sample, local distribution more concentrated |
| Tasks become closer in feature space | $H_{\text{task}} \downarrow$ | Task correlation increases, reducing diversity |
| Tasks become more separated in feature space | $H_{\text{task}} \uparrow$ | Task differences expand feature space coverage |

We now analyze how the kernel-entropy estimator behaves when the dataset is modified. Recall that $H_{\text{task}}$ is computed as

$$H_{\text{task}} = -\frac{1}{|\mathcal{X}|} \sum_{\mathbf{x}_i \in \mathcal{X}} \log \hat{p}(\mathbf{x}_i), \qquad \hat{p}(\mathbf{x}_i) = \frac{1}{|\mathcal{X}|} \sum_{j=1}^{|\mathcal{X}|} K_\sigma(\mathbf{x}_i, \mathbf{x}_j),$$

where $\hat{p}(\mathbf{x}_i)$ is the Parzen window density estimate at $\mathbf{x}_i$. Intuitively, when $\mathbf{x}_i$ lies in a dense region, $\hat{p}(\mathbf{x}_i)$ is high and $\log \hat{p}(\mathbf{x}_i)$ becomes less negative, contributing less to the entropy. Thus, adding points that further increase local density will lower $H_{\text{task}}$, while adding points that cover previously sparse regions will raise it.

Concretely, adding a new sample from a previously unseen task that lies far from existing points increases the coverage of the feature space and results in a higher $H_{\text{task}}$. By contrast, adding more samples from the same task that are tightly clustered around existing points increases local density but introduces little new information, thereby reducing $H_{\text{task}}$. Similarly, when different tasks become

closer to each other in the feature space, their distributions start to overlap, making the overall feature distribution more concentrated and lowering the entropy. Conversely, when tasks are pushed farther apart, the overall distribution becomes more spread out, and $H_{\text{task}}$ increases.

Finally, removing samples can have two opposite effects depending on which samples are removed: removing outliers that lie far from all tasks decreases the feature coverage and reduces $H_{\text{task}}$, while removing redundant clustered samples can slightly increase $H_{\text{task}}$ by making the density estimate less over-concentrated.

Overall, $H_{\text{task}}$ captures not just the number of samples but also their arrangement in the feature space: simply duplicating data points can actually reduce the measured diversity, whereas adding complementary samples that populate underrepresented regions increases it.

# E   MORE DETAILS ABOUT $L_{\text{DATASET}}$

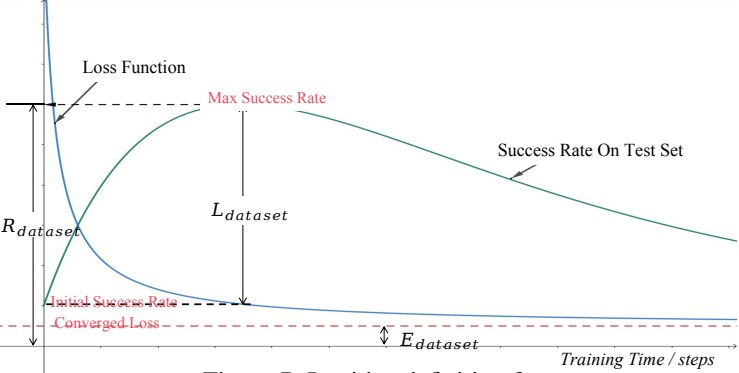

Figure 7: Intuitive definition for

## E.1   MEMORY FACTOR $E_t$

Consider the $E_t$ of a task:

$$E_t = \frac{1}{\log_{1p}(\bar{L}_t)} \mathbb{E}_{\mathbf{x}_{t,i}, \mathbf{x}_{t,j} \in \mathcal{X}_t}[K_\sigma(\mathbf{x}_{t,i}, \mathbf{x}_{t,j})],$$

where $\mathbf{x}_{t,i} \in \mathcal{X}_t \subset \mathbb{R}^D$ denotes samples from task $t$, and $K_\sigma(\mathbf{x}_{t,i}, \mathbf{x}_{t,j}) \in (0, 1]$ is a kernel measuring similarity, with self-similarity $K_\sigma(\mathbf{x}_{t,i}, \mathbf{x}_{t,i}) = 1$.

- **Highly overlapping samples:** If samples in the task are nearly identical, then $K_\sigma(\mathbf{x}_{t,i}, \mathbf{x}_{t,j}) \approx 1$ for all $i, j$. Consequently, the average kernel $\mathbb{E}[K_\sigma(\mathbf{x}_{t,i}, \mathbf{x}_{t,j})] \approx 1$, and $E_t$ is large, indicating that the task is easy to overfit.

- **Highly dispersed samples:** If samples are far apart in feature space, then $K_\sigma(\mathbf{x}_{t,i}, \mathbf{x}_{t,j}) \approx 0$ for $i \neq j$, leaving only the diagonal self-similarities equal to 1. The average kernel becomes $\mathbb{E}[K_\sigma(\mathbf{x}_{t,i}, \mathbf{x}_{t,j})] \approx 1/N_t$, yielding a smaller $E_t$, indicating a task that is difficult to overfit.

In summary, $E_t$ directly reflects the intra-task sample concentration: the more concentrated the samples, the easier it is for a model to memorize them; the more dispersed, the harder it is to overfit.

This formulation captures both intra-task properties, inter-task transfer effects, and the relative representation of tasks in the dataset, providing a comprehensive measure of dataset-level learnability.

## E.2   EXPRESSIVENESS FACTOR $R_t$

The expressiveness factor $R_t$ quantifies the potential of a task to expose the model to diverse patterns. It consists of two components: *directional coverage* $H(X_t)$ and *spatial coverage* $C(X_t)$:

$$R_t = H(X_t) \cdot C(X_t).$$

**Step 1: Construct the sample matrix** Let $X_t \in \mathbb{R}^{N_t \times D}$ be the feature matrix of task $t$, where each row $\mathbf{x}_{t,i} \in \mathbb{R}^D$ represents a sample's $D$-dimensional feature vector.

**Step 2: Compute the covariance matrix** The covariance matrix $\Sigma_t \in \mathbb{R}^{D \times D}$ captures the linear correlations between feature dimensions:

$$\Sigma_t = \frac{1}{N_t - 1}(X_t - \bar{X}_t)^\top (X_t - \bar{X}_t),$$

where $\bar{X}_t = \frac{1}{N_t} \sum_{i=1}^{N_t} \mathbf{x}_{t,i}$ is the sample mean vector of task $t$. Intuitively, $\Sigma_t$ tells us how much each feature dimension varies and how feature dimensions co-vary.

**Step 3: Eigen-decomposition** Compute the eigenvalues $\lambda_1, \ldots, \lambda_D$ of $\Sigma_t$:

$$\Sigma_t \mathbf{v}_i = \lambda_i \mathbf{v}_i, \quad i = 1, \ldots, D,$$

where $\mathbf{v}_i$ are the eigenvectors. - Each eigenvalue $\lambda_i$ measures the variance along the corresponding principal direction $\mathbf{v}_i$. - Large $\lambda_i$ means the data spreads widely along that direction; small $\lambda_i$ indicates low variability.

**Step 4: Directional coverage $H(X_t)$** Directional coverage is the entropy of the normalized eigenvalues:

$$H(X_t) = -\sum_{i=1}^{D} \frac{\lambda_i}{\sum_j \lambda_j} \log \frac{\lambda_i}{\sum_j \lambda_j}.$$

- If variance is concentrated in a few dimensions, $H(X_t)$ is small. - If variance is uniformly distributed across dimensions (isotropic data), $H(X_t)$ is maximized at $\log D$.

**Step 5: Spatial coverage $C(X_t)$** Spatial coverage measures the effective spread of the samples:

$$C(X_t) = N_t \cdot \tanh\left(\frac{\bar{d}_t}{\sigma}\right), \quad \bar{d}_t = \frac{2}{N_t(N_t - 1)} \sum_{i<j} \|\mathbf{x}_{t,i} - \mathbf{x}_{t,j}\|,$$

where $\bar{d}_t$ is the average pairwise distance and $\sigma$ scales the nonlinearity. - Dense, clustered samples $\to$ small $C(X_t)$. - Widely spread samples $\to C(X_t)$ approaches $N_t$.

**Step 6: Combining components** Finally, the expressiveness factor:

$$R_t = H(X_t) \cdot C(X_t),$$

captures both high-dimensional variability and spatial spread.

**Extreme cases for $R_t$**

- **Highly clustered / low-rank samples:** $\lambda_1 \approx \sum_j \lambda_j$, $\bar{d}_t \approx 0 \Rightarrow R_t \approx 0$.

- **Isotropic / well-spread samples:** $\lambda_i \approx \lambda_j$ for all $i, j$, $\bar{d}_t$ large $\Rightarrow R_t \approx (\log D) \cdot N_t$.

This detailed decomposition shows exactly how the covariance structure of the data determines task expressiveness.

**Interpretation** Thus, $R_t$ captures the potential for generalization: tightly clustered or low-dimensional samples yield small $R_t$, while isotropic and widely spread samples yield large $R_t$. Together with $E_t$, the raw task learnability

$$L_{t,\text{raw}} = R_t^\beta \cdot E_t^{1-\beta}$$

inherits these bounds, providing a principled measure of task difficulty and pattern richness.

### E.3 DATASET LEARNABILITY

After computing task-level learnability $L_{t,\text{raw}}$, we consider the overall contribution of each task to the dataset-level learnability by accounting for two key factors: the task proportion $\pi_t$ and inter-task influence $I_{it}$.

**Task Proportion** $\pi_t$    The proportion of a task in the dataset reflects its relative representation:

$$\pi_t = \tanh\left(\frac{|X_t|}{\sum_{t'} N_{t'} \cdot \sigma_{\text{model}}}\right),$$

where $|X_t|$ is the number of samples in task $t$, and $\sigma_{\text{model}}$ controls the scaling according to the model's capacity.

- **Extreme low proportion:** If $|X_t| \to 0$, then $\pi_t \to 0$, meaning that the task is effectively ignored in $L_{\text{dataset}}$.
- **High proportion:** If $|X_t|$ is large relative to other tasks, $\pi_t \to 1$, giving full weight to the task's learnability.

**Inter-task Influence** $I_{it}$    We quantify the knowledge transfer from other tasks using a Gaussian kernel over task centers:

$$I_{it} = K_\sigma(\mu_i, \mu_t), \quad \mu_t = \frac{1}{|X_t|} \sum_{\mathbf{x} \in X_t} \mathbf{x}.$$

- Tasks with similar centers ($\mu_i \approx \mu_t$) have $I_{it} \approx 1$, indicating strong positive transfer.
- Tasks that are very different ($\mu_i$ far from $\mu_t$) have $I_{it} \approx 0$, contributing little to $L_{t,\text{adjusted}}$.

**Adjusted Task Learnability**    Combining $\pi_t$ and $I_{it}$, the adjusted learnability for task $t$ is

$$L_{t,\text{adjusted}} = \pi_t \cdot \sum_i I_{it} L_{i,\text{raw}}.$$

This formulation ensures that:

- Tasks with small representation are down-weighted, avoiding overestimation.
- Knowledge transfer from similar tasks can positively influence $L_{t,\text{adjusted}}$.
- Extreme cases:
  1. $\pi_t = 0 \Rightarrow L_{t,\text{adjusted}} = 0$.
  2. $I_{it} = 0$ for all $i \neq t \Rightarrow L_{t,\text{adjusted}} = \pi_t L_{t,\text{raw}}$.
  3. High $\pi_t$ and strong $I_{it}$ maximize $L_{t,\text{adjusted}}$.

**Dataset Learnability**    Finally, the overall dataset learnability is the mean over all tasks:

$$L_{\text{dataset}} = \frac{1}{|T|} \sum_{t \in T} L_{t,\text{adjusted}}.$$

This measure captures both intra-task properties (via $L_{t,\text{raw}}$), task representation (via $\pi_t$), and

### E.4   BEHAVIOR ANALYSIS

| Sample Change | Affected Factor(s) | Explanation |
|---|---|---|
| Add a sample to a different task | $\pi_t$ | Reduces relative proportion of task $t$, other factors unchanged |
| Add a sample to task $t$, not tightly clustered | $\pi_t, R_t, E_t$ | $\pi_t$ increases; $R_t$ increases due to higher local entropy; $E_t$ decreases due to lower density |
| Add a sample to task $t$, tightly clustered | $\pi_t, R_t, E_t$ | $\pi_t$ increases; $R_t$ decreases due to reduced entropy; $E_t$ increases due to higher density |

Intuitively, $L_{\text{dataset}}$ reflects three key aspects:

1. **Intra-task properties:** Each task's raw learnability $L_{t,\text{raw}}$ encodes its expressiveness and ease of memorization, as discussed in previous sections.
2. **Inter-task knowledge transfer:** $I_{it}$ allows tasks to benefit from related tasks, increasing $L_{t,\text{adjusted}}$ when similar tasks exist.
3. **Task representation:** $\pi_t$ ensures that tasks with low representation are appropriately down-weighted, while highly represented tasks dominate the dataset-level score.

**Illustrative Cases**   The influence of adding new samples can be interpreted as follows:

- **Adding a sample to a different task:** The relative proportion $\pi_t$ of task $t$ decreases, as the new sample increases the size of another task. Other factors, such as $R_t$ and $E_t$, remain unchanged for task $t$, leading to a slight reduction in its adjusted learnability.

- **Adding a dispersed sample to task $t$:** The task proportion $\pi_t$ increases slightly. The expressiveness $R_t$ also increases, as the added sample expands the spatial coverage and variability of the task. Simultaneously, the ease-of-memorization $E_t$ decreases, because the lower density of dispersed samples makes memorization harder.

- **Adding a tightly clustered sample to task $t$:** The task proportion $\pi_t$ increases, reflecting the larger sample size. However, $R_t$ decreases due to the reduced variability and lower spatial coverage caused by the tight cluster. The ease-of-memorization $E_t$ increases, because the dense cluster makes memorization easier.

Overall, $L_{\text{dataset}}$ provides a principled measure that balances intra-task learnability, inter-task transfer, and task representation, capturing how well a dataset enables a model to learn and generalize across tasks.

## F   EXPERIMENTAL RAW DATA

| Goal | | Object | | Spatial | | Ten | | Goal-10 (Part 1) | | Goal-10 (Part 2) | |
|------|------|------|------|------|------|------|------|------|------|------|------|
| GT | Pred | GT | Pred | GT | Pred | GT | Pred | GT | Pred | GT | Pred |
| 40.00 | 0.176855 | 88.00 | 0.174945 | 94.00 | 0.184274 | 52.00 | 0.163141 | 58.00 | 0.175880 | 52.00 | 0.162242 |
| 52.00 | 0.171998 | 70.00 | 0.176867 | 96.00 | 0.178297 | 34.00 | 0.168488 | 52.00 | 0.171049 | 52.00 | 0.167559 |
| 90.00 | 0.175622 | 90.00 | 0.175374 | 76.00 | 0.177805 | 48.00 | 0.163538 | 82.00 | 0.174654 | 58.00 | 0.162636 |
| 82.00 | 0.182041 | 84.00 | 0.174876 | 96.00 | 0.176177 | 20.00 | 0.161562 | 84.00 | 0.181037 | 28.00 | 0.160671 |
| 96.00 | 0.182089 | 72.00 | 0.177211 | 82.00 | 0.178632 | 62.00 | 0.169810 | 94.00 | 0.181085 | 70.00 | 0.168874 |
| 96.00 | 0.183188 | 86.00 | 0.176035 | 84.00 | 0.177643 | 58.00 | 0.168100 | 98.00 | 0.182178 | 54.00 | 0.167173 |
| 78.00 | 0.182173 | 88.00 | 0.179833 | 88.00 | 0.181925 | 66.00 | 0.169925 | 64.00 | 0.181168 | 78.00 | 0.168988 |
| 40.00 | 0.173624 | 94.00 | 0.178839 | 90.00 | 0.180233 | 44.00 | 0.165051 | 38.00 | 0.172666 | 44.00 | 0.164140 |
| 88.00 | 0.185266 | 90.00 | 0.179673 | 80.00 | 0.176968 | 38.00 | 0.166796 | 92.00 | 0.184244 | 56.00 | 0.165876 |
| 94.00 | 0.187556 | 86.00 | 0.178222 | 72.00 | 0.177289 | 68.00 | 0.169851 | 98.00 | 0.186522 | 64.00 | 0.168914 |

Table 5: Ground truth (GT) and predicted values (Pred) for all datasets.

## G   ADDITIONAL DETAILS ON EXPERIMENT

In this appendix, we provide further discussion on the rationale behind our experimental design, including dataset selection and the fine-tuning approach.

**Challenges with Large-scale Pretraining**   Training a large embodied model from scratch or pre-training on a large-scale dataset is prohibitively time- and resource-intensive Kim et al. (2024). Such an approach would require significant computational resources and long training times, making it impractical for large-scale validation across multiple datasets.

**Limitations of Existing Real-world Datasets**   Using existing real-world datasets introduces an-other challenge: evaluating success rates on individual tasks is often infeasible because the original task scenarios cannot be exactly replicated. This limitation prevents reliable dataset-level evaluation and reduces the number of ground-truth points available for validating our learnability metric.

**Advantages of Simulated Datasets and Fine-tuning**   To overcome these challenges, we focus on simulated datasets based on RoboSuite Zhu et al. (2020), which allow precise replication of tasks and environments. We adopt a fine-tuning paradigm rather than full pretraining, which enables efficient

measurement of the effect of different datasets on model performance. Additionally, since dataset-level learnability can be defined as the average of task-level learnabilities, task-level evaluation provides a larger number of ground-truth points and a finer-grained assessment, facilitating a more reliable validation of our proposed metric.

# H    REAL ROBOT EXPERIMENTS SET UP DETAIL

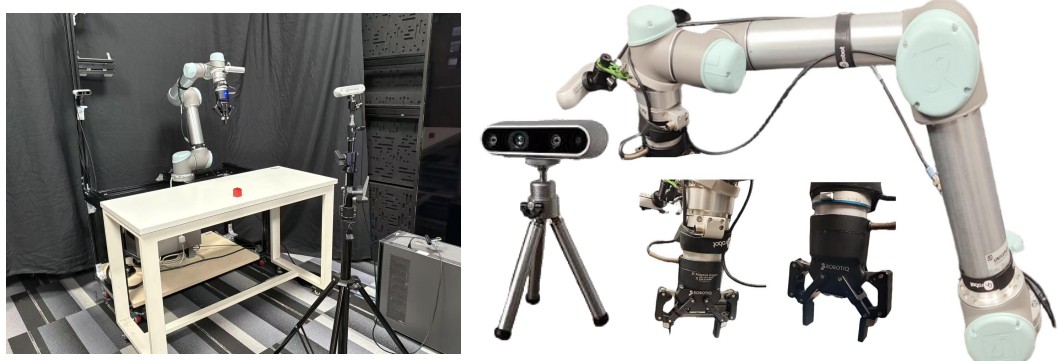

Figure 8: Real machine experimental scenes and experimental facilities

During the real-robot validation, we use a UR5 robotic arm equipped with a Robotiq 2F-140 gripper. Intel RealSense-D455 cameras are used for observation: positioned in front of the workspace, as shown in Figure 8. All experiments are conducted in a controlled in-lab environment.

The procedure consists of two phases:

1. **Data Collection and Zero-shot Evaluation:** We first collect demonstration data and evaluate the zero-shot success rate using OpenVLA.

2. **Training and Evaluation:** After training the model on the collected dataset, we re-evaluate the success rate on the same tasks.

We consider two datasets with different characteristics:

**Dataset 1: Comprehensive Coverage**    For the first dataset, the cubes on the table are distributed to cover all possible locations, ensuring that the sampling covers the full workspace. The control trajectories are deliberate and minimal, without redundancy, with clear and explicit goals.

**Dataset 2: Sparse Coverage**    For the second dataset, samples are collected only from a small subset of the workspace, leading to limited coverage. The control trajectories are hesitant and exploratory, reflecting less certainty and more variation in the demonstrations.

To make our evaluation transparent, we provide a simple case study illustrating how we define success and failure (Fig. 9). The first row shows a successful trial where the robot successfully grasps the cube, which we count as a success. The second row shows a trial where a collision occurred and triggered an emergency stop, which we count as a failure. The third row shows a completely failed trial where the robot attempts to grasp in an incorrect location, also counted as a failure.

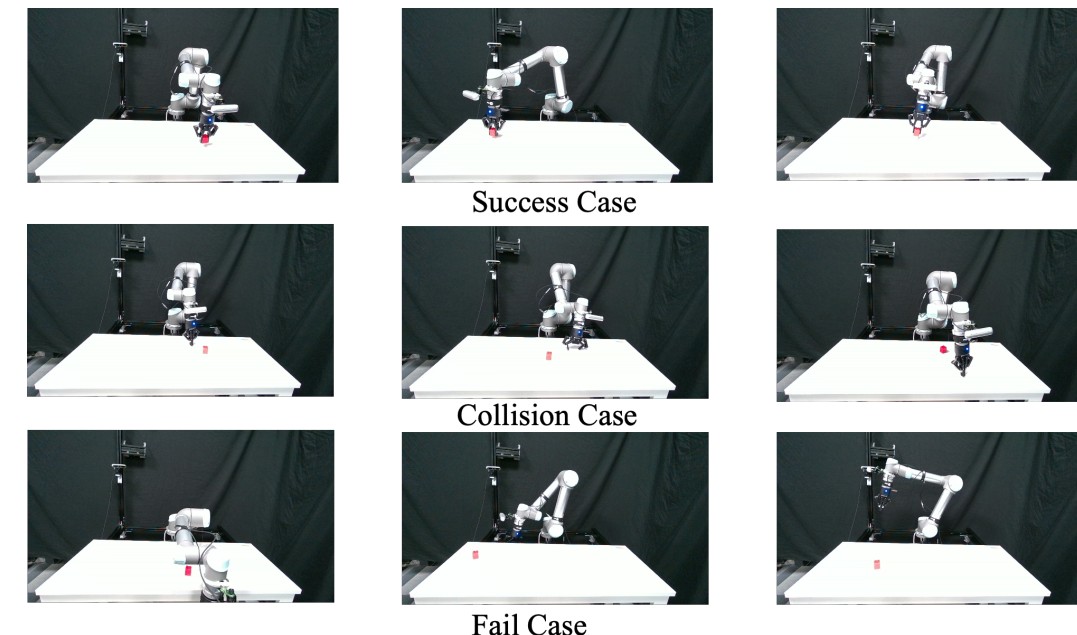

Figure 9: Case Study for real-bot evaluation, showing three cases

# I HUMAN VS. ALGORITHM

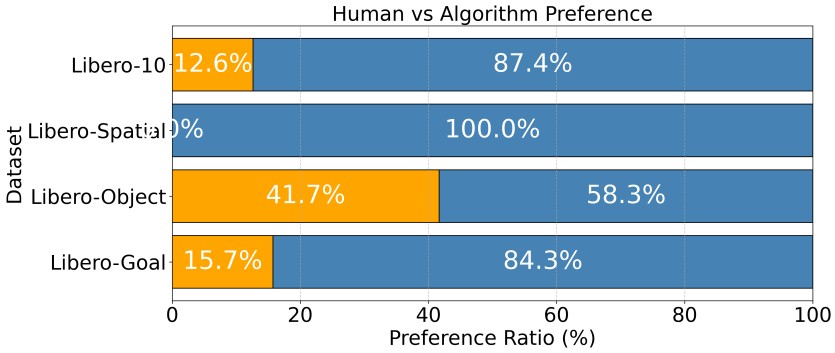

Figure 10: Human vs Algorithm. Orange: Human, Blue: Algorithm

To better understand how humans perceive dataset learnability, we conducted a small-scale user study on the four Libero datasets. Participants were asked to rate the **learnability** of each dataset based on their subjective impression of task difficulty and sample distribution, without access to any model performance data. We observed that human ratings were largely driven by two intuitive factors: (1) the perceived difficulty of the underlying tasks, and (2) the rough number of available samples. Participants rarely examined the detailed structure of the datasets (e.g., diversity of samples, trajectory coverage), which may lead to underestimation or overestimation of learnability in certain cases.

Figure 10 compares the correlation between human subjective scores and ground-truth task success rates against our algorithmic measure of dataset learnability. As shown, our approach consistently achieves much higher correlation across all metrics (SROCC, KROCC, PLCC), suggesting that our method captures dataset-level learnability more faithfully than human intuition.

## J    RELATED EMBODIED DATASET INTRODUCTION

**ASU Table Top:** UR5 performing table-top pick/place/rotate tasks. Each step includes RGB observation (224×224), robot state (7D), joint velocities, and optional language instruction (with 512-D embedding). Actions are 7D continuous vectors (joint velocities + gripper + terminate signal). Dataset contains 110 training episodes (737.6 MiB).

**Austin Buds:** Franka stylized kitchen tasks. Each step includes main camera and wrist camera RGB observations (128×128), a 24D robot state (joint angles, gripper position, end-effector pose), and a natural language instruction with 512-D embedding. Actions are 7D continuous vectors (6D end-effector delta pose + 1D gripper position). Dataset contains 50 training episodes (1.49 GiB).

**Austin Sailor:** Franka tablesetting tasks. Each step includes main and wrist camera RGB observations (128×128), a 25D robot state (default state 8D, end-effector 16D, gripper 1D), and a natural language instruction with 512-D embedding. Actions are 7D continuous vectors (3D ee relative pos + 3D ee relative rotation + 1D gripper). Dataset contains 240 training episodes (18.85 GiB).

**BC-Z:** Robot performing pick/place/rotation tasks. Each step includes a downsampled camera image (171×213×3), the robot's current state (end-effector axis-angle 3D, position 3D, gripper 1D), episode success (0-1), sequence length, and a natural language instruction with 512-D embedding. Actions are recorded as the next 10 steps: each with 3D position delta, 3D rotation delta (axis-angle), and gripper target (10 future steps). Episodes include DAgger labels for autonomous vs. teleoperator actions. Dataset contains 9746 episodes (size unknown).

**Berkeley Cable Routing:** Routing a cable into clamps on a table top. Each step includes four RGB images (main, top, wrist225, wrist45), a 7D robot state, and a natural language instruction with 512-D embedding. Actions are 7D vectors: 3D rotation delta, 3D world vector, 1D terminate episode. Dataset contains 1482 training episodes and 165 test episodes (4.67 GiB).

**Berkeley Fanuc Manipulation:** Fanuc robot performing various manipulation tasks. Each step includes main and wrist camera RGB observations (224×224), a 13D robot joint state (6 joint angles, 1 gripper open, 6 joint velocities), a 7D end-effector state (x, y, z + 4x quaternion), and a natural language instruction with 512-D embedding. Actions are 6D continuous vectors (dx, dy, dz + droll, dpitch, dyaw). Dataset contains 415 training episodes (8.85 GiB).

**NYU Franka Play:** Franka robot interacting with toy kitchens. Each step includes right and left RGB observations (128×128), right and left depth images (128×128), 13D robot state (7 joint angles, 3 EE xyz, 3 EE rpy), and a 512-D language embedding of the instruction. Actions are 15D continuous vectors (7 joint velocities, 3 EE delta xyz, 3 EE delta rpy, 1 gripper, 1 terminate). Dataset contains 365 training episodes and 91 validation episodes (5.18 GiB).

**Jaco Play:** Jaco 2 robot performing pick and place on table top. Each step includes wrist and main RGB images (224×224), 7D end-effector Cartesian position, 6D end-effector velocity, 8D joint positions, and a 512-D language embedding of the instruction. Actions consist of 7D continuous vectors (1 gripper closedness, 3 terminate episode, 3 world velocity). Dataset contains 976 training episodes and 109 test episodes (9.24 GiB).

**NYU Door Opening:** Robot performing cabinet, microwave, and door opening tasks. Each step includes a 720×960 RGB image and a 512-D language embedding of the instruction. Actions are 7D continuous vectors consisting of gripper closedness, 3D rotation delta, terminate episode, and 3D world velocity. Dataset contains 435 training episodes and 49 test episodes (7.12 GiB).

**Roboturk:** Real robot dataset with cloth folding and bowl stacking tasks. Each step includes a 480×640 RGB image and a 512-D language embedding of the instruction. Actions are 7D continuous vectors consisting of gripper closedness, 3D rotation delta, terminate episode, and 3D world velocity. Dataset contains 1,796 training episodes and 199 test episodes (45.39 GiB).

**Taco Play:** Franka arm interacting with kitchen objects. Observations include 150×200 static RGB, 84×84 gripper RGB, depth maps, 512-D language embeddings, and structured instructions. Actions are 7D absolute/relative gripper poses. Dataset contains 3,242 training episodes and 361 test episodes (47.77 GiB).

**Toto:** Franka arm performing scooping and pouring tasks. Observations include 480×640 RGB images, 512-D language embeddings, and 7-D robot joint states. Actions include gripper open/close, 3-D rotation delta, and 3-D world vector velocity. Dataset splits and total size are unspecified.

**Viola:** Franka robot interacting with stylized kitchen tasks. Observations include 224×224 RGB images from workspace and in-hand cameras, 512-D language embeddings, end-effector pose (16D), joint states (7D), and gripper width (1D). Actions include 3D rotation delta, gripper closedness, world vector, and terminate episode signal. Dataset split: 135 train, 15 test. Total size: 10.40 GiB.

## K    PSEUDOCODEAND ALGORITHM TIME COMPLEXITY

### K.1    PSEUDOCODE OF DIVERSITY AND LEARNABILITY ESTIMATORS

---

**Algorithm 1:** Compute Task Diversity Entropy $H_{\text{task}}$

---

**Input:** Feature matrix $X \in \mathbb{R}^{N \times D}$, kernel bandwidth $\sigma$ (optional)
**Output:** Task diversity entropy $H_{\text{task}}$
1 **Step 1:** Compute pairwise Euclidean distances:   $d_{ij} \leftarrow \|x_i - x_j\|_2$ for all $i, j = 1, \ldots, N$
2 **Step 2:** If $\sigma$ is not provided:   Use the median of upper-triangular distances:
 $\sigma \leftarrow \text{median}\{d_{ij} \mid i < j\}$
3 **Step 3:** Compute kernel similarity matrix:   $K_{ij} \leftarrow \exp\left(-\frac{d_{ij}^2}{2\sigma^2}\right)$
4 **Step 4:** For each sample $i$, compute local density:   $p_i \leftarrow \frac{1}{N} \sum_{j=1}^{N} K_{ij}$
5 **Step 5:** Estimate entropy:   $H_{\text{task}} \leftarrow -\frac{1}{N} \sum_{i=1}^{N} \log(p_i + \varepsilon)$
6 **Return** $H_{\text{task}}$

---

**Algorithm 2:** Compute Learning Ease $L_{\text{dataset}}$ with Task Transfer

---

**Input:** Feature matrix $X$, task labels $y$, task lengths $\{T_t\}$, dataset name $d$, trade-off $\beta$, kernel
   bandwidth $\sigma$
**Output:** Learning ease $L_{\text{dataset}}$ and per-task ease $\{L_t\}$
1 **Initialize:** $L_t^{raw} \leftarrow 0$ for all tasks $t$
2 **Step 1:** For each task $t \in \text{unique}(y)$:
   1. Extract task subset $X_t = \{x_i \mid y_i = t\}$ with $N_t = |X_t|$
   2. Compute pairwise distances $d_{ij}$ within $X_t$
   3. Compute similarity matrix: $S_{ij} \leftarrow \exp(-d_{ij}^2/2\sigma^2)$
   4. Normalize: $P_{ij} = S_{ij}/\sum_j S_{ij}$
   5. Compute local entropy: $h_t \leftarrow -\frac{1}{N_t} \sum_i \sum_j P_{ij} \log(P_{ij} + \varepsilon)$
   6. Compute average pairwise distance: $d_{avg}^t \leftarrow \text{mean}\{d_{ij} \mid i < j\}$
   7. Compute covariance entropy $R_t \leftarrow H(\text{eigvals}(\text{Cov}(X_t))) \cdot \tanh(d_{avg}^t/\sigma)$
   8. Compute expected density $E_t \leftarrow \frac{1}{N_t} \sum_i \frac{\sum_j S_{ij}}{\log(1+T_t)}$
   9. Get task prior $\pi_t$ from dataset-specific ratio and apply scaling: $\pi_t \leftarrow \tanh(\pi_t/c)$
   10. Combine: $L_t^{raw} \leftarrow (R_t^{\beta}) \cdot (E_t^{1-\beta})$
**Step 2:** Compute task centers:   $c_t \leftarrow \frac{1}{N_t} \sum_{x \in X_t} x$
**Step 3:** Compute inter-task similarity:   $S_{task}^{ij} \leftarrow \exp\left(-\frac{\|c_i - c_j\|_2^2}{2\sigma_c^2}\right)$
**Step 4:** Adjust $L_t$ by task transfer:   $L_t^{adj} \leftarrow \pi_t \cdot \sum_j S_{task}^{ij} \cdot L_j^{raw}$
**Step 5:** Aggregate dataset-level ease:   $L_{\text{dataset}} \leftarrow \frac{1}{|\mathcal{T}|} \sum_t L_t^{adj}$
**Return** $L_{\text{dataset}}, \{L_t^{adj}\}$

---

### K.2 COMPUTATIONAL COMPLEXITY ANALYSIS

**Complexity of $\hat{H}_{\mathbf{data}}$**    Algorithm 1 requires computing a kernel value between each pair of samples $(\mathbf{x}_i, \mathbf{x}_j)$, resulting in $n^2$ evaluations in total. Each kernel computation involves an $O(d)$ operation (where $d$ is feature dimension). Thus, the overall complexity is

$$\mathcal{O}(n^2 d)$$

which is quadratic in dataset size. This cost becomes significant for very large datasets. In practice, several approximations can reduce the cost:

- **Subsampling:** Estimate $\hat{H}_{\mathrm{data}}$ on a random subset of samples.
- **Kernel truncation:** Discard kernel contributions when $\|\mathbf{x}_i - \mathbf{x}_j\| > \tau$ (negligible density).
- **Approximate nearest neighbors:** Restrict the summation to $k \ll n$ nearest neighbors.

**Complexity of $L_{\mathbf{data}}$**    Algorithm 2 requires finding $k$-nearest neighbors for each sample. A naive implementation performs pairwise distance computation in $O(n^2 d)$, followed by partial sorting $O(nk \log n)$. The total complexity is

$$\mathcal{O}(n^2 d + nk \log n).$$

For large $n$, this is dominated by the $O(n^2 d)$ distance computation. In practice, efficient data structures such as KD-trees or approximate nearest neighbor (ANN) libraries (e.g., FAISS, HNSW) reduce complexity to approximately $O(n \log n)$ query time, making $L_{\mathrm{data}}$ scalable to datasets with millions of samples.

**Memory Complexity**    Both algorithms require storing either the full $n \times n$ kernel matrix ($O(n^2)$ memory) or on-the-fly computation with $O(nd)$ memory for the features. For large-scale experiments, we adopt block-wise computation to keep memory usage within GPU/CPU limits.

**Summary**    Both metrics are *computationally tractable* for medium-sized datasets ($n < 10^5$) and can be further accelerated using random subsampling and approximate nearest neighbor search, which we verify to yield nearly identical metric values in practice (less than $1\%$ relative error). For extremely large datasets, a practical alternative is to use $L_{t,\mathrm{raw}}$ as a surrogate for $L_{t,\mathrm{transfer}}$, which avoids expensive transfer computing while still providing a meaningful estimate of task learnability.

## L    LOW-LEVEL DIVERSITY

This section analyzes five low-level visual statistics across 21 embodied datasets. For each dataset, we randomly sampled images from all tasks and computed the following five measures to quantify their basic visual variability. The results are shown in Fig. 11.

COMPUTATION OF EACH FACTOR

Each sampled image $\mathbf{I}$ is first converted into its grayscale image $\mathbf{I}_g$ and HSV representation. Then, we compute the following five metrics:

- **Lightness** ($L$,Luminance): Measured as the mean pixel intensity of the grayscale image:

$$L = \frac{1}{|\mathbf{I}_g|} \sum_{p \in \mathbf{I}_g} \mathbf{I}_g(p).$$

- **Structural Information** ($\sigma$,Spatial Infomation): Defined as the standard deviation of grayscale intensities:

$$\sigma = \sqrt{\frac{1}{|\mathbf{I}_g|} \sum_p \left(\mathbf{I}_g(p) - L\right)^2},$$

which reflects the amount of contrast structure in the image.

- **Contrast** ($C$)**:** Following the local contrast definition, we divide the image into $4 \times 4$ grids and compute the average of $(\max - \min)$ differences in each grid, normalized by $\sigma$:

$$C = \frac{1}{16 \cdot \max(\sigma, 1)} \sum_{k=1}^{16} \big( \max(\mathbf{I}_g^{(k)}) - \min(\mathbf{I}_g^{(k)}) \big).$$

- **Colorfulness** ($\mathcal{C}$, Chrominance): Based on Hasler and Süsstrunk's metric, we compute

$$\mathcal{C} = \sqrt{\sigma_{rg}^2 + \sigma_{yb}^2} + 0.3\sqrt{\mu_{rg}^2 + \mu_{yb}^2},$$

where $rg = R - G$ and $yb = 0.5(R + G) - B$ are opponent color components.

- **Blur** ($B$)**:** Measured as the variance of image gradients, obtained via a $3 \times 3$ Sobel operator:

$$B = \mathrm{Var}\big(\nabla_x \mathbf{I}_g + \nabla_y \mathbf{I}_g\big).$$

A lower value indicates stronger blurring.

Together, these metrics provide a comprehensive characterization of low-level visual diversity, capturing brightness distribution, structural variation, contrast richness, colorfulness, and image sharpness.

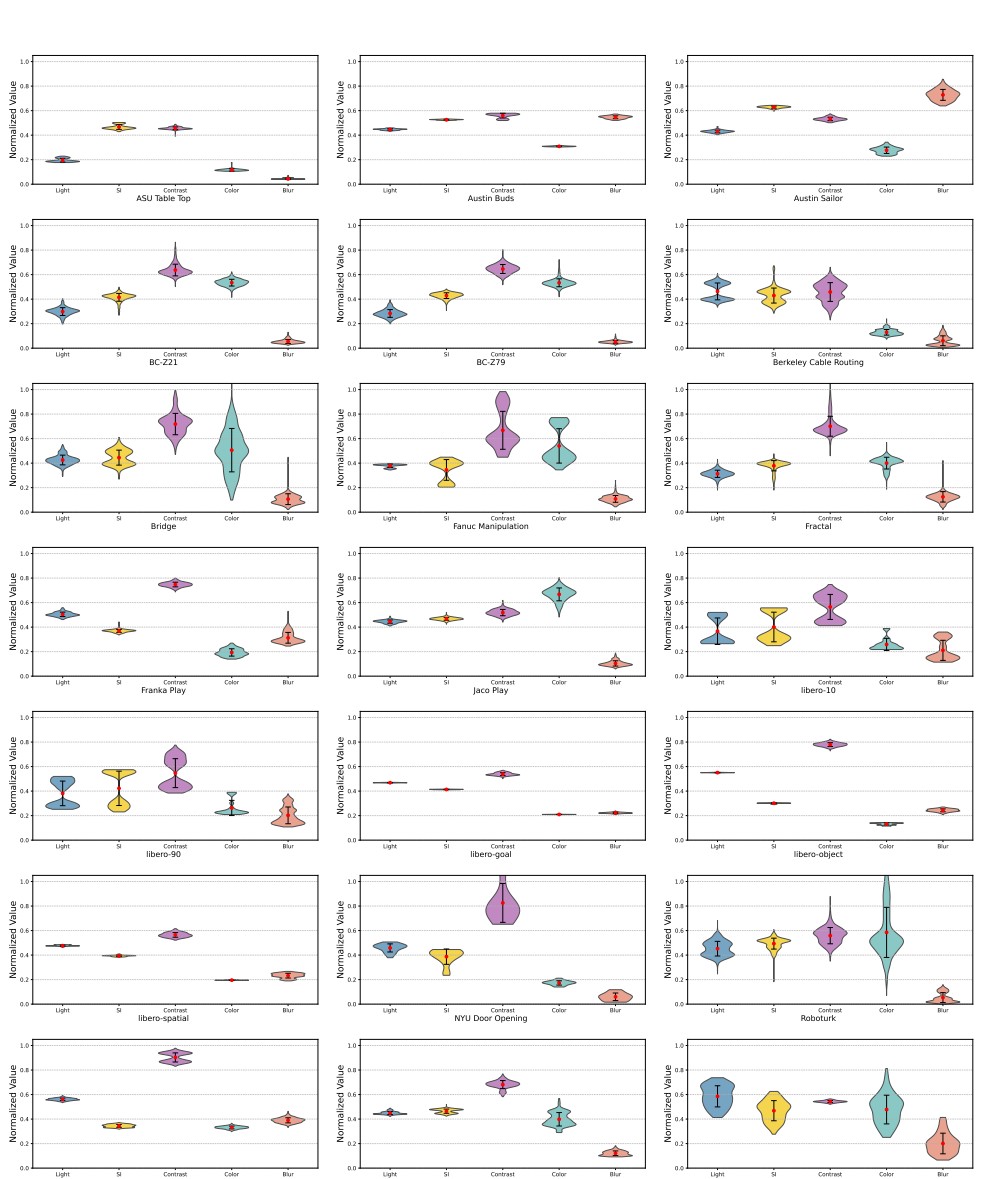

Figure 11: Low-level Diversity for 21 Embodied Datasets.

