# OpenReview forum: "Data Assessment for Embodied Intelligence"
_ICLR.cc/2026/Conference — Submitted to ICLR 2026_

### Official Review · Reviewer_KKhd · 2025-11-01

**Soundness:** 2
**Presentation:** 4
**Contribution:** 2
**Rating:** 2
**Confidence:** 4

**Summary:**

For embodied datasets, the authors created multimodal representations that are used to quantify the amount of information within a dataset, while also developing an algorithm to assess its learnability without requiring training.

**Strengths:**

1. The paper tries to come up with ways of quantifying the utility of embodied data which is an important problem to adress. Also some of this techniques could also be used to understand datasets from other domains.

2. The authors of the paper seem to have a done a thorough comparison with the related works and the introduction and the related works section seem to be well done.

**Weaknesses:**

1. Some of the design choices in the paper appear somewhat arbitrary (see questions for details).

2. It is also unclear how strongly the dataset score correlates with the accuracy results (see questions for details).

3. Some of the claims made in the paper may be insufficiently substantiated (see questions for details).

**Questions:**

1. Could you clarify why only three frames are selected? Wouldn’t it make more sense to process the videos using a dedicated video model (for example, Video-LLaVA: https://huggingface.co/docs/transformers/en/model_doc/video_llava)? Additionally, could you explain why longer videos are penalized, as shown in Equation 2? If a video consists of repeated or constant frames, it shouldn’t necessarily be harder to memorize simply because it’s longer. Furthermore, this approach might fail to capture the diversity of tasks or actions within a single video. Hence is there an implicit assumption on the number of tasks per video?

2. In the conclusion the authors say: "In this work, we presented two principled, data-driven tools for assessing embodied datasets. First, we introduced diversity entropy, characterizing the information richness of a dataset. Experiments on 21 large-scale embodied datasets, we observe that simply scaling the number of episodes yields diminishing returns in information gain". However, I'm not sure if I saw such an experiment. Where was this experiment conducted?

3. Could you clarify how Table 5 (Appendix F) should be interpreted? Specifically, what do “ground truth” and “predicted” refer to in this context? If I’m understanding it correctly, wouldn’t this suggest that the metric and the accuracy are not correlated?

4. Could you explain why the learnability of one model would be influenced by the representations from CLIP? How can the learning dynamics of one model be meaningfully related to the representations of another?

5. Why is the model’s learnability represented as a single scalar value (Equation 5)? What does this number signify conceptually, and how is it actually computed?

---

### Official Review · Reviewer_VCAi · 2025-11-01

**Soundness:** 3
**Presentation:** 3
**Contribution:** 3
**Rating:** 6
**Confidence:** 2

**Summary:**

This paper introduces two data driven tools for embodied datasets, a continuous metric called diversity entropy to quantify the information content of the dataset and an algorithm to estimate dataset learnability. The authors demonstrate the effectiveness of their framework on both simulation and real-world datasets.

**Strengths:**

This work introduces a means to evaluate the quality of the datasets that are central to learning based methods in embodied agents. This is very crucial since the agent performance depends on these datasets. I believe this work will provide insights for future work.

**Weaknesses:**

Overall the paper is well written and easy to understand. I didn’t find any significant weaknesses. However, I have a few clarifying questions listed below.

**Questions:**

I have a few clarifying questions for the authors:

1. Since CLIP is not trained on robot trajectories, how sensitive do you think are your diversity and learnability estimates to the choice of encoder? Have you tried pretrained encoders trained on embodied datasets?
2. How did you choose the sigma for H_data in eq. 1? How sensitive is H_data to this choice?
3. The metric seems to be reflective of visual variance. Did you test whether H_data correlates with semantic diversity, such as object or task variety?

---

### Official Review · Reviewer_rkwR · 2025-11-07

**Soundness:** 3
**Presentation:** 3
**Contribution:** 3
**Rating:** 4
**Confidence:** 3

**Summary:**

The author proposed two algorithms for data valuation in multimodal embodied datasets. The first method, named diversity entropy, is calculated based on a unified multimodal representation for each data. This quantifies information content of the data. The second method utilizes two factors, ease of memorization and expressiveness, to measure the learnability of the data. The authors validate these metrics on simulated and real-world datasets, analyzing 21 popular embodied datasets totaling over 800GB.

**Strengths:**

- While data valuation has been an established area, I really like the author's systematic approach of dividing valuation into diversity and learnability, and analyze the two holistically to give a better picture of the quality of the data. The topic of data valuation is also relevant in building VLA foundation models, as VLA training data is often noisy and focus on a narrow range of tasks.
- The diversity entropy based on Parzen window estimation and the learnability factors are rigorously defined
- The valuation metrics are strongly correlated with downstream performance.

**Weaknesses:**

- From my understanding, the contribution is primarily on developing a new data valuation method. If this is the case, the author should compare the method to other data valuation methods in the related works section. Currently, the related works section only covers Embodied datasets and VLA Models, which are not very useful to understand the contribution of the work.
- While the abstract claims the model use "unified multimodal representation", this representation is in fact just video frames, as described in Sec. 3.1. It would make the work more interesting if the representation can leverage information across multiple modalities, as I'm not entirely convinced different modalities represent same information.
- While authors test the method on multiple datasets, they only used one model (OpenVLA-7B) to validate model's downstream performance. It's not clear whether the performance/correlation transfers to other models.
- Figure 6 caption: multimodle -> multimodal. Please proofread for other spelling mistakes.

**Questions:**

See weaknesses. I would consider raising my score if the author could provide a comprehensive comparison against other works in data valuation domain, and evaluate the method on models other than OpenVLA.

---

### Official Review · Reviewer_N6CJ · 2025-11-08

**Soundness:** 2
**Presentation:** 2
**Contribution:** 2
**Rating:** 2
**Confidence:** 4

**Summary:**

This paper tackles the underexplored problem of quantitatively evaluating the diversity and learnability of embodied datasets. The authors propose two data-driven, training-free metrics: (1) Diversity Entropy, a continuous measure of dataset information richness derived from unified multimodal representations, and (2) a Learnability Estimator, an interpretable algorithm that predicts how easily a model can learn from a dataset without training. The authors validate their approach on 21 simulated and real-world embodied datasets, including Libero, BridgeData, and Fractal, and show strong correlations between predicted and empirically measured learnability. The study suggests that dataset diversity and learnability can be meaningfully assessed without costly model retraining.

**Strengths:**

- The authors provide the first quantitative and interpretable metrics for assessing embodied dataset diversity and learnability without model retraining.
- They demonstrate strong empirical validation across diverse datasets, suggesting robustness.

**Weaknesses:**

- The reliance on CLIP as a universal multimodal encoder limits generality; alternative embeddings (e.g., OpenVLA latent space) could yield different results.
- The proposed metrics are heuristic approximations, not theoretically guaranteed proxies for model learnability.
- Real-world validations are limited (two UR5 datasets); broader experimental diversity would strengthen claims.
- Diversity entropy depends heavily on bandwidth and kernel choice, yet sensitivity analysis is missing.
- The learnability estimator, while interpretable, introduces many hyperparameters (β, σ_model, σ_center) that lack principled tuning or uncertainty reporting.
- The discussion underplays potential biases in CLIP-derived latent spaces and how they may distort entropy measurements.

**Questions:**

- How sensitive are diversity entropy and learnability metrics to the CLIP feature space choice or visual encoder type?
- Could the authors compare their metrics to mutual information or intrinsic dimension estimators as alternative diversity measures?
- How would the proposed framework scale to video-based embodied datasets exceeding terabyte scale?
- Is the learnability estimator applicable to language-conditioned or 3D embodied tasks, where temporal semantics differ?
- Can the authors provide a reproducible toolkit or open-source implementation for community use?

**Details Of Ethics Concerns:**

No ethics concerns.

---

### Meta-Review · Area_Chair_JzCK · 2026-01-04

**Summary:**

This paper  aims to address an important and timely problem, training-free evaluation of embodied datasets, and proposes an interpretable framework that distinguishes dataset diversity from learnability.  The authors propose (1) Diversity Entropy, a continuous entropy-based measure computed from unified visual representations of trajectories, and (2) a Learnability Estimator that decomposes dataset utility into memorization ease and expressiveness, aiming to predict downstream learning gains without retraining models. The motivation of reducing reliance on costly retraining procedures is compelling.

However, the majority of reviewers (3 out of 4) recommend rejection, and no author response or feedback has been provided.

**Reviewer Concerns:**

Three reviewers (N6CJ, KKhd, rkwR) raise serious concerns regarding the methodological foundations and generality of the proposed metrics:

- the heavy reliance on CLIP-based visual representations as a proxy for multimodal embodied data,

- the lack of principled justification and robustness analysis for key hyperparameters,

- the interpretability and validity of compressing dataset learnability into a single scalar.

Collectively, these issues impact the perceived soundness and generalizability of the approach and go beyond presentation-level concerns. **No author response or clarification was provided to address these points.**

**Reviewer Scores:**

Scores: 2，4，6，2

The majority of reviewers (3 out of 4) recommend rejection. Reviewer VCAi provides a more favorable assessment with low confidence (score: 6, confidence: 2). I concur with the overall assessment of the reviewers.

---

### Decision · Program_Chairs · 2026-01-26

Reject